# Quantum clocks and the temporal localisability of events in the presence of gravitating quantum systems

Esteban Castro-Ruiz[1,2,3 ✉], Flaminia Giacomini[1,2,4], Alessio Belenchia [5] & Časlav Brukner[1,2]

The standard formulation of quantum theory relies on a fixed space-time metric determining the localisation and causal order of events. In general relativity, the metric is influenced by matter, and is expected to become indefinite when matter behaves quantum mechanically. Here, we develop a framework to operationally define events and their localisation with respect to a quantum clock reference frame, also in the presence of gravitating quantum systems. We find that, when clocks interact gravitationally, the time localisability of events becomes relative, depending on the reference frame. This relativity ia a signature of an indefinite metric, where events can occur in an indefinite causal order. Even if the metric is indefinite, for any event we can find a reference frame where local quantum operations take their standard unitary dilation form. This form is preserved when changing clock reference frames, yielding physics covariant with respect to quantum reference frame transformations.

[1] Faculty of Physics, Vienna Center for Quantum Science and Technology (VCQ), University of Vienna, Boltzmanngasse 5, A-1090 Vienna, Austria. [2] Institute for Quantum Optics and Quantum Information (IQOQI), Austrian Academy of Sciences, Boltzmanngasse 3, A-1090 Vienna, Austria. [3] Centre for Quantum Information and Communication, École polytechnique de Bruxelles, Université libre de Bruxelles, CP 165, 1050 Brussels, Belgium. [4] Perimeter Institute for Theoretical Physics, 31 Caroline St. N, Waterloo, ON N2L 2Y5, Canada. [5] Centre for Theoretical Atomic, Molecular, and Optical Physics, School of Mathematics and Physics, Queen's University, Belfast BT7 1NN, UK. ✉email: esteban.castro.ruiz@univie.ac.at

Quantum theory allows us to predict the probabilities of obtaining certain outcomes when we perform operations on a physical system. These operations are specified by laboratory procedures, including preparations, transformations and measurements. Apart from these operational elements, quantum theory relies on a definite space-time metric to make empirical predictions. Indeed, the metric is often implicit in the definition of an operation, and it determines the causal structure of the space-time where the operations are performed.

According to general relativity, the space-time metric is obtained by solving Einstein's equations. Generically, a solution to these equations depends on the matter distribution, which is assumed to be classical. Understanding the consequences of replacing gravitating classical matter by gravitating quantum systems in general relativity is at the heart of the problem of quantum gravity[1,2]. At the moment, a fully satisfactory and broadly accepted theory of quantum gravity is lacking, and it is far from clear how, if at all, the essential notions of quantum mechanics and general relativity will be modified in the more fundamental theory[3,4].

In the absence of such a theory, making quantum mechanical predictions when quantum systems act as gravitational sources is an important challenge. This is because gravitating quantum systems can lead to an indefinite spacetime metric, that is, a metric whose values are not given a priori, independently of any observations carried out on quantum systems. This gives rise to well known difficulties[5–11], which include the following: (i) The dynamical law of quantum theory (the Schrödinger equation) relies on a time parameter to specify the evolution of quantum systems. What plays the role of such time parameter in the absence of a definite metric? (ii) If the space-time metric is indefinite, the causal order between different operations is also indefinite. In this case, how are we supposed to apply the quantum mechanical rules for calculating probabilities for measurements corresponding to a set of observables? (iii) The physical realisation of an operation on a quantum system typically relies on "background" degrees of freedom, which are crucial for defining the operation. For example, the pointer position of a clock can define the time when the operation is applied[12]. If the metric field is indefinite, the clock does not "know" how fast to tick, due to an uncertainty in its time dilation[13]. (More precisely, the clock gets entangled with the gravitating degrees of freedom.) In this situation, how are we supposed to define the operation in the first place?

In this work, we propose a method for tackling some aspects of the above difficulties. We develop a framework for "time reference frames", which are (quantum) reference frames associated to quantum clocks. (Recently, clocks at the interplay between quantum mechanics and general relativity have gained significant attention[14–19]). The following paragraph summarises our findings in the context of the difficulties mentioned above:

Regarding (i), we further develop a "timeless" approach[20–24] according to which time emerges though correlations between "what a clock shows" and the state of the system. We consider a set of multiple clocks and describe the evolution of a quantum system according to individual clocks from the set. In particular, we consider cases in which the space-time metric is indefinite due to gravitating quantum systems in a superposition of energy or position eigenstates. We show that a notion of unitary time evolution arises with respect to each of these clocks, even in the presence of gravitating quantum systems. Moreover, the Schrödinger equation is covariant (form invariant) under the transformation from one time reference frame to another. This fact constitutes a concrete implementation of the covariance of physical laws in quantum reference frames advocated for in ref. [25].

With respect to (ii), we introduce a local, operational definition of an "event", in which an operation is applied to a quantum system conditioned on a clock reading a specific "time", and for which we can use the timeless approach to calculate its probability of occurrence even in cases when the order between events is not definite. Moreover, we study the temporal localisation of events with respect to different time reference frames, and find that, when quantum clocks interact with gravitating quantum systems, the temporal localisability of an event becomes relative, depending on the time reference frame. This result provides a concrete physical realisation, in terms of quantum systems which are sources of the gravitational field, of the claim reported in refs. [26,27], that the localisability of events is observer-dependent, and that operations might be performed in time-delocalised subsystems. We argue that this relativity characterises an indefinite metric, which can lead to an indefinite causal structure of events, where whether a given event is in the causal past, causal future or is causally disconnected from another event is not a factual property of the world. We illustrate this fact by using our framework to analyse the gravitational quantum switch[12]. In view of (iii), one might naively expect that the time evolution with respect to a gravitationally interacting clock A is "noisy" or decoherent. After all, A is expected to get entangled due to its interaction, and its quantum state is expected to become mixed. We show, however, that this view is (quantum) reference frame dependent, and one can find a transformation to the time reference frame of A with respect to which time evolution is unitary. This means, in particular, that if we define an "event" by applying an operation to a quantum system when clock A shows a specific "time", this operation will be represented, in A's time reference frame, by its unitary dilation, like in ordinary quantum mechanics with a fixed metric. Therefore, by "jumping" into a suitable time reference frame, quantum operations take their usual form, even if the metric is indefinite. This result resonates with the concept of the "quantum equivalence principle" advocated by Hardy[9,10], and can be seen as another example of covariance of physical laws in quantum reference frames[25], in a case where the space-time metric is indefinite.

## Results

**Reference frames for events and time evolution**. The central idea of this work is that of a time reference frame. By definition, a time reference frame is a quantum temporal reference frame associated to a quantum clock. An observer A uses a time reference frame to define the temporal localisation of events and describe the physics of a system S as it evolves in time, as defined by the quantum clock. By events we mean any quantum operation performed on S. When there is no room for confusion, we will use the same symbol to denote the observer and the clock. To formulate this idea, we consider as a starting point the timeless approach to quantum mechanics[20,21,28], in particular the version of ref. [21]. In "Methods (Review of the timeless approach to quantum mechanics)" we present summary of the latter formulation (see also refs. [23,24] for similar approaches). Mathematically, the evolution of S in the time reference frame of A is encoded in the history state

$$|\Psi\rangle = \int dt |t\rangle_A \otimes |\psi(t)\rangle_S. \qquad (1)$$

Here, $|t\rangle_A$ is an eigenstate of the time operator $\hat{T}_A$, $\hat{T}_A|t\rangle_A = t|t\rangle_A$, and corresponds to clock A showing time $t$. The state $|\psi(t)\rangle_S$, called the reduced state, has the physical interpretation of "the state of system S when clock A shows time $t$". According to the timeless formulation, $|\Psi\rangle$ is subjected to a

constraint

$$\hat{C}|\Psi\rangle = 0, \qquad (2)$$

from which we obtain $|\Psi\rangle$.

We are interested in comparing the temporal localisation of events and the time evolution of a system as "seen" by different time reference frames. For this reason, we consider an extension of the timeless formulation to include more than one clock, say A and B, and a system of interest, S. In our framework, operations on S define events, and each clock labels the temporal localisation of these events and assigns a different time evolution operator to S and the rest of the clocks. In "Methods (Changing time reference frames)" we develop a general framework to change from the evolution operator with respect to A to that with respect to B. Our treatment is compatible with the fact that, in special and general relativity, time is not absolute but rather is defined by the reading of a clock moving along a specific world-line.

Despite our approach being motivated by relativistic physics, it departs from the usual case of special and general relativity, because we do not assume that the space-time metric is fixed. Since a space-time is defined by a manifold and a metric, our clocks and systems are not, strictly speaking, embedded in space-time. Nevertheless, we insist on the operational definition of events with respect to quantum clocks, associating to each clock a reference frame by which we will define "evolution" and "temporal localisation".

The operational meaning of our framework is depicted in Fig. 1 and further explained in "Methods (Operational meaning of the framework)". Mathematically, by expressing the history state $|\Psi\rangle$ in either the reference frame of A or B, we will be able to track the temporal localisation of events with respect to each reference frame. Although in this paper we focus on the temporal localisation of events, it is natural to assume that A and B have access to an additional set of spatial degrees of freedom, which allows them to operationally localise events in space as well. In the present setup, the inclusion of such (quantum) degrees of freedom can be carried out by the methods developed in ref. [25]. In the remaining of this work, whenever we mention the notion of space or distance, we will do so having this context in mind.

**Non interacting clocks.** Let us illustrate the above ideas by means of a simple example. Imagine that A and B have two clocks, which interact neither with each other nor with anything else in the experiment. The time measured by each clock corresponds to an operator $\hat{T}_I$, for $I = A, B$. We assume that A and B are perfect clocks: $[\hat{T}_I, \hat{H}_I] = i$, where $\hat{H}_I$ is the Hamiltonian of the clock I, for $I = A, B$. (see "Methods (Review of the timeless approach to quantum mechanics)"). Suppose that A "sets up" an event by means of an interaction between S and her ancilla, a. The event is produced when A's clock is in a sharp state of $\hat{T}_A$ with eigenvalue $t_A^* > 0$. A similar setting holds for B, with a corresponding time $t_B^* > 0$ and ancilla b. Under these conditions, the constraint equation describing the experiment is

$$(\hat{H}_A + \hat{H}_B + \hat{f}_A(\hat{T}_A) + \hat{f}_B(\hat{T}_B))|\Psi\rangle = 0. \qquad (3)$$

Here, $\hat{f}_A(\hat{T}_A) = \delta(\hat{T}_A - t_A^*)\hat{K}^{(A)}$, where $\hat{K}^{(A)}$ is a hermitian operator on $\mathcal{H}_S \otimes \mathcal{H}_a$, the Hilbert space of the system and A's ancilla. A similar statement holds for $\hat{f}_B(\hat{T}_B) = \delta(\hat{T}_B - t_B^*)\hat{K}^{(B)}$ and the ancilla b. For simplicity of notation, we have left the tensor products implicit in Eq. (3) and written, for example, $\hat{H}_A \otimes \mathbb{1}_R$ simply as $\hat{H}_A$ (Here, R (the "rest") denotes all the systems that are not clocks, namely S, the system of interest, and the ancillas, a and b). We follow this notation throughout the paper. We have assumed that the free Hamiltonian for the system is trivial,

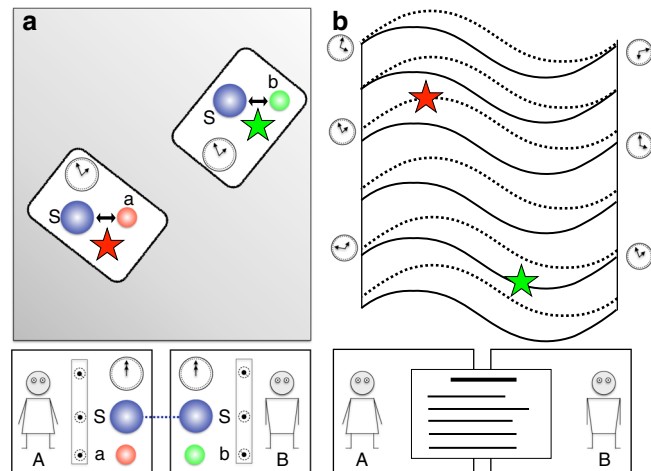

**Fig. 1 Description of the operational meaning of the framework.** A and B perform experiments on S in two stages. In the preparation stage (**a**), A and B specify the states of their clocks, subsystems (depicted by a blue ball) and ancillas (depicted by a red ball, a for A and by a green ball, b for B). They do so by freely choosing the knob settings that control these systems. S can be a composite system entangled between A and B. This is indicated by the line joining A and B's subsystems. A and B can program their clocks so that an interaction between S and the ancillas, a and b, is turned on at a specific local time, say $t^*$. In the detection stage (**b**), A and B read the measurement results by looking at the outcome of their clocks and ancillas. By assumption, the choices made by A and B in the preparation stage do not depend on the results obtained in the detection stage, even if the experiments take place in an indefinite causal structure. As the figure shows, we assume that both parties have access to all the data of the experiment. With these data, A and B "map" the set of events into "space-time", which we depict as a manifold foliated by surfaces of constant time according to A (dotted lines) and according to B (solid lines). The event produced by A (B) is depicted by a red (green) star. In the case depicted here, the two events are sharply localised in both A and B's time reference frames. This feature is consistent with a fixed space-time background, where the time localisation of events is absolute. However, in this work we show that there are situations involving gravitating quantum systems, which lead to an indefinite metric background. In such backgrounds, whether an event is sharply localisable in time or not depends on the time reference frame.

$\hat{H}_S = 0$. In this paper we focus on the case where the events are triggered when the state of the clocks are sharply peaked around a given time. However, more general models are possible by suitably choosing the hermitian-operator-valued functions $\hat{f}_I(\hat{T}_I)$, for $I = A, B$.

The history state in the time reference frame of A reads (see Supplementary Notes 1 and 2)

$$|\Psi\rangle = \int dt_A \, |t_A\rangle_A \otimes \, e^{-it_A\hat{H}_B} \, \mathrm{T} \, e^{-i\int_0^{t_A} ds(\hat{f}_A(s) + \hat{f}_B(s + \hat{T}_B))} |\psi_A(0)\rangle_{\bar{A}}, \qquad (4)$$

where T denotes the usual time ordering operator, defined by $\mathrm{T}\hat{f}(s_1)\hat{f}(s_2) = \Theta(s_2 - s_1)\hat{f}(s_2)\hat{f}(s_1) + \Theta(s_1 - s_2)\hat{f}(s_1)\hat{f}(s_2)$, for any operator-valued function $\hat{f}$ of s. $\Theta(s)$ is the usual Heaviside function, equal to 1 if $s > 0$, 1/2 if $s = 0$, and 0 otherwise. $|\psi_A(0)\rangle_{\bar{A}} = \int dt'_B \, \varphi(t'_B) \, |t'_B\rangle_B \otimes |\chi\rangle_R$ is the state of all systems, except clock A, conditioned on clock A being in the state $|t = 0\rangle_A$. In this work, $\bar{I}$, for $I = A, B, \ldots$, denotes all subsystems except for subsystem I. We use primed time variables to refer to the initial state of the clocks. It is physically meaningful to assume that the support of the wave packet

of clock B does not overlap with the time defining any of the events. In Supplementary Note 2 we explicitly construct a wave-packet with this feature.

The state $\left|\psi_A(t_A)\right\rangle_{\bar{A}} = \langle t|_A \cdot |\Psi\rangle$ represents the state of all systems except clock A, conditioned on clock A being in the state $|t\rangle_A$. Because Eq. (4) contains all such conditional states for each $t_A$, the history state $|\Psi\rangle$ contains all the information regarding the time evolution and measurements made on system $\bar{A}$. Importantly, the history state represented in the time reference frame of A describes the subsystem $\bar{A}$ evolving unitarily from the initial state $\left|\psi_A(0)\right\rangle_{\bar{A}}$, with the evolution operator $\hat{U}(t_A)_{\bar{A}} = exp(-it_A\hat{H}_B) \; T \; exp(-i\int_0^{t_A} ds \; (\hat{f}_A(s) + \hat{f}_B(s + \hat{T}_B)))$. By unitarity, the normalisation of $\left|\psi_A(0)\right\rangle_{\bar{A}}$ is preserved in $t_A$.

In Eq. (4), the argument of $\hat{f}_A$ is a c-number. This means that, in A's time reference frame, the time at which operation A is applied is always sharp —it happens at the well-defined time $t_A^*$. By contrast, the argument of $\hat{f}_B$ in Eq. (4) depends on the operator $\hat{T}_B$. When $\hat{f}_B(\hat{T}_B)$ acts on the initial state $\left|\psi_A(0)\right\rangle_{\bar{A}}$, the argument of $\hat{f}_B$ becomes dependent on $t_B'$, and is therefore modulated by the wave packet $\varphi(t_B')$. Assuming that $\varphi(t_B')$ is not sharply peaked but rather has a finite width $\sigma$, this effect will lead to an uncertainty, from the point of view of A, as to when the operation triggered by clock B is applied. The larger the width $\sigma$, the greater the uncertainty. This effect is easily seen if we write down explicitly the conditional state $\left|\psi_A(t_A)\right\rangle_{\bar{A}} = \langle t_A|_A \cdot |\Psi\rangle$. For simplicity, suppose that $t_B^* < t_A < t_A^*$, so that we do not need to discuss the operation triggered by clock A, which we already know to be sharply localised in A's time reference frame. A simple calculation yields,

$$\left|\psi_A(t_A)\right\rangle_{\bar{A}} = \int dt_B' \; \varphi(t_B') \; T \; e^{-i\int_0^{t_A} ds \hat{f}_B(s+t_B')} \left|t_A + t_B'\right\rangle_B \otimes |\chi\rangle_R. \tag{5}$$

Equation (5) is nothing but a coherent superposition of quantum states, each of them depending on $t_B'$. The amplitudes of the superposition are given by $\varphi(t_B')$. For each of these amplitudes, the (operator valued) function $\hat{F}_B(t_A, t_B') := \int_0^{t_A} ds \; \hat{f}_B(s + t_B') = \Theta(t_A + t_B' - t_B^*)\hat{K}^{(B)}$ takes different values. In A's reference frame, B's operation will already be applied at a given time $t_A$ only for those amplitudes corresponding to a $t_B'$ such that $t_A + t_B' > t_B^*$. Because, given a time $t_A$, B's operation will already be applied only for some amplitudes, this analysis clearly shows that the time localisation of B's operation is uncertain with respect to A. In conclusion, A's description of the experiment features two events: one is sharply defined at time $t_A^*$, and the other one, which A describes as triggered by B's clock, is uncertain, due to the uncertainty of B's clock (see Fig. 2).

How does the experiment look like from the point of view of B? As can be seen from the equations derived in "Methods (Changing time reference frames)", the history state in B's time reference frame is exactly the same as that in A's, provided that the wave-packet $\varphi(t_B')$ is symmetric under the operation $t \longrightarrow -t$. This is not surprising, given the symmetry of Eq. (3). The important point is that B describes the initial state of A's clock by the same wave packet $\varphi(t_A')$. As a consequence of the finite wave-packet width, $\sigma$, the localisation of A's operation in time, as defined by B, will have an uncertainty modulated by $\varphi(t_A')$ (see Fig. 2b). In contrast, the operation triggered by clock B will be always sharp in B's time reference frame.

To summarise, from the point of view of a given time reference frame I, a measurement triggered by clock I is always localised in time, while the measurements triggered by the other clocks are, in

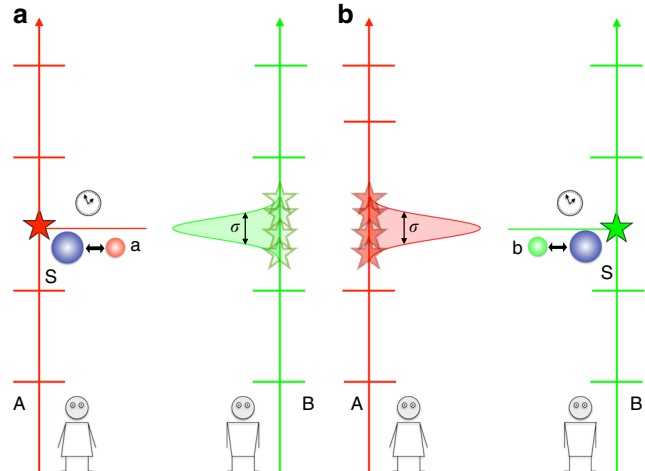

**Fig. 2 Relative localisability of events for non interacting clocks with unsharp initial states.** This figure describes the history state $|\Psi\rangle$, solution to Eq. (3), in the time reference frame of A (**a**) and of B (**b**). There are two events, (depicted by a red and a green star), taking place in the experiment. In the event depicted by a red star, the system S, (depicted by the blue ball), interacts with A's ancilla, a (depicted by a red ball). In the event depicted by a green star, the system S interacts with the ancilla b (depicted by a green ball), initially under the control of B. In A's time reference frame (**a**), the event depicted by a red star is sharply localised in time. In contrast, the event depicted by a green star has an uncertainty, characterised by $\sigma$, in its time localisation, due to the "fuzzy" initial state of clock B in A's time reference frame (see main text). The roles are reversed in the time reference frame of B (**b**). In this frame, it is the event depicted by a green star which is sharply localised in time (as defined by clock B), whereas the event depicted by a red star exhibits some time uncertainty in B's reference frame. This uncertainty is due to the fact that, in B's frame, the initial state of clock A is "fuzzy".

general, delocalised with respect to $t_I$, the local time of clock I. As we will see in the following, this relativity of localisation in time is not only a feature of a "poorly-chosen" initial state. Rather, it is an unavoidable effect when clocks experience time dilation due to their interaction with gravitating quantum systems.

**Evolution with respect to gravitationally interacting clocks.** In the previous subsection we have seen how to describe the dynamical evolution and events of a quantum experiment with respect to different quantum clocks, A and B. In our analysis, we made the important assumption that the clocks do not interact with each other. However, practical reasons aside, this assumption must fundamentally break down once the gravitational effects of A and B become significant—gravity is universal and cannot be shielded. Furthermore, the situation becomes all the more radical when we consider the quantum properties of A and B: Any clock, say B, must run in a superposition of different energies of its Hamiltonian, $\hat{H}_B$[29]. Because energy-momentum determines the metric field via Einstein's equations, each of these energies corresponds to a different metric background. Therefore, the fact that the state of B is indefinite with respect to the observable $\hat{H}_B$ means that the metric background, determined by B, is indefinite, too[13]. In this situation, how would another clock, say A, describe the time evolution of a quantum experiment?

Here we aim to answer the above question from the point of view of time reference frames. We will show that, even in the lack of a definite space-time background, an operational notion of time evolution can be defined if we "jump" into the time reference frame of a given clock, say A. Importantly, A can interact

gravitationally and get entangled with another clock, say B, from the point of view of yet another observer, C. Because the state of A is mixed in C's frame, one may naively think that the time evolution with respect to A is "noisy" or decoherent. However, we will show that this needs not be the case. We will present a model where, even if C "sees" clock A entangled with clock B due to their gravitational interaction, a consistent notion of "local time" exists in the time reference frame of A, with respect to which time evolution is unitary. This fact is closely related to the result reported in ref. [25], that entanglement and superposition are (quantum) reference frame dependent features.

Let clocks A, B and C be subjected to the constraint

$$\left(\sum_I \hat{H}_I + \frac{1}{2}\sum_{I \neq J} \lambda_{IJ}\hat{H}_I\hat{H}_J\right)|\Psi\rangle = 0. \tag{6}$$

Here, $\lambda_{IJ} = -G/(c^4 x_{IJ})$, where $G$ is the gravitational constant, $c$ is the speed of light, and $x_{IJ}$ is the relative distance between clocks I and J. The indices take the values I, J = A, B and C. Eq. (6) describes, to the first order in $1/c^2$, clocks interacting with each other via their gravitational field. Note that, in principle, the relative distances between the clocks, $x_{IJ}$, (and therefore the $\lambda_{IJ}$s) are quantum operators as well. However, in this work we assume, for simplicity, that the $x_{IJ}$s are c-numbers and that they are time independent (with respect to any clock). This assumption means that the clocks follow semiclassical trajectories and remain at the same relative distance with respect to each other. Under this condition, the $\lambda_{IJ}$s are c-numbers as well. (This coupling between quantum clocks was introduced in[13] and was studied in a timeless context in ref. [16]).

Let us analyse how time evolution emerges from the point of view of C. In order to do this, we act on Eq. (6) with $\langle\tau|_C$. We obtain

$$i\left(1 + \lambda_{AC}\hat{H}_A + \lambda_{BC}\hat{H}_B\right)\frac{d}{d\tau}|\psi(\tau)\rangle_{\bar{C}} = \left(\hat{H}_A + \hat{H}_B + \lambda_{AB}\hat{H}_A\hat{H}_B\right)|\psi(\tau)\rangle_{\bar{C}}. \tag{7}$$

Note that Eq. (7) is not of the form of the Schrödinger equation, and is not the description of time evolution with respect to the proper time of clock C, due to the extra term $i(\lambda_{AC}\hat{H}_A + \lambda_{BC}\hat{H}_B)\partial_\tau$ on the left hand side. Before we complete the "jump" into C's time reference frame, it is interesting to discuss heuristically the physical meaning of Eq. (7). Consider first the simpler situation of a single quantum system evolving on a fixed space-time background given by the weak field metric $ds^2 = -(1 + 2\Phi(x)/c^2)c^2dt^2 + dx \cdot dx$. For a static observer, the state of the system $|\psi\rangle$ evolves under the Schrödinger equation $i\partial_t|\psi\rangle = \sqrt{-g_{00}}H_{rest}|\psi\rangle$, where $g_{00} = -(1 + 2\Phi(x)/c^2)$ and $H_{rest}$ is the Hamiltonian in the reference frame where the system is at rest[30,31]. Note that, for this observer $\sqrt{-g_{00}} = \dot{\tau}$, where $\tau$ denotes the proper time of the system and the dot denotes derivative with respect to time $t$. Defining $H_{lab.} = \sqrt{-g_{00}}H_{rest}$ as the Hamiltonian in the laboratory reference frame, and using the fact that, for a static observer, $\partial_t = \dot{\tau}\partial_\tau$, we can rewrite the evolution of the system as

$$i\sqrt{-g_{00}}\frac{d}{d\tau}|\psi\rangle = H_{lab.}|\psi\rangle. \tag{8}$$

Under the approximation $\sqrt{-g_{00}} \approx 1 + \Phi(x)/c^2$, we see that Eq. (8) is precisely of the form of Eq. (7), for the case where the gravitational potential $\Phi$ is sourced by a gravitating classical body. We can therefore interpret Eq. (7) as a generalisation of Eq. (8) for the case where the "gravitational potential" is sourced by a gravitating quantum system. With this intuition, the operator in brackets on the left-hand side of Eq. (7) can be interpreted as a "redshift" operator, which, together with the (proper) time

derivative operator, forms a "$d/d\hat{t}$" operator with respect to (an operator-valued) coordinate time $\hat{t}$. Then, explicitly,

$$\frac{d}{d\hat{t}} := \left(1 + \lambda_{AC}\hat{H}_A + \lambda_{BC}\hat{H}_B\right)\frac{d}{d\tau}. \tag{9}$$

(For a rigorous definition of a derivative with respect to an operator, see ref. [32].) This interpretation is tightly related to the idea of a quantum reference frame[25], where the set of symmetry transformations is generalised to include reference frames associated to quantum systems. Such transformations are obtained by promoting the parameter associated to the transformation to an operator acting on an additional Hilbert space. In "Methods (Gravitational quantum switch)" and "Methods (Remarks on coordinates for gravitating quantum systems)" we further discuss and analyse the concept of an operator-valued, quantum coordinate time and its transformations in the context of time reference frames.

In order to complete the process of moving into C's time reference frame, we formally divide by the redshift factor operator in Eq. (7), with the assumption that the energies of the state of the system C are small enough such that no divergences occur. In "Methods (Remarks on coordinates for gravitating quantum systems)", we analyse physically the conditions under which divergences do not appear. Under these assumptions, we obtain the Schrödinger Equation in the time reference frame of C:

$$i\frac{d}{d\tau}|\psi(\tau)\rangle_{\bar{C}} = \frac{\hat{H}_A + \hat{H}_B + \lambda_{AB}\hat{H}_A\hat{H}_B}{1 + \lambda_{AC}\hat{H}_A + \lambda_{BC}\hat{H}_B}|\psi(\tau)\rangle_{\bar{C}}. \tag{10}$$

Eq. (10) shows that, in the time reference frame of C, the evolution of clocks A and B is unitary, despite the fact that there is a non-negligible interaction term between C and the clocks A and B in Eq. (6). Importantly, in C's reference frame, there is an interaction between A and B, leading in general to entanglement between A and B in the view of C. In order to make this point clearer, let us assume that all $\lambda_{IJ}$s are small. We obtain, to first order in $\lambda_{IJ}$,

$$i\frac{d}{d\tau}|\psi(\tau)\rangle_{\bar{C}} = \left(\tilde{H}_A + \tilde{H}_B + \tilde{\lambda}_{AB}\tilde{H}_A\tilde{H}_B\right)|\psi(\tau)\rangle_{\bar{C}}, \tag{11}$$

where $\tilde{H}_I = \hat{H}_I(1 - \lambda_{IC}\hat{H}_I)$, for I = A, B, and $\tilde{\lambda}_{AB} = \lambda_{AB} - \lambda_{AC} - \lambda_{BC}$. Eq. (11) is precisely the Schrödinger equation for two gravitationally interacting clocks, A and B, to the first order in $1/c^2$. Note that, because C is not infinitely far-away from A and B, the Hamiltonians of A and B in C's frame are blue-shifted from the original Hamiltonians appearing in Eq. (6). In C's frame, the gravitational coupling between A and B in Eq. (11) is shifted as well.

Nothing in principle prevents us from deriving an analogous equation from the perspective of either A or B. Suppose we were to "jump" into the time reference frame of A. Clearly, under similar assumptions as for the case of C, we would obtain an evolution equation of the form of Eq. (10) (or Eq. (11)) but with the indices A and C interchanged. Because Eq. (10) is a Schrödinger equation, the evolution of B and C in A's time reference frame is also unitary, even if A gets entangled with B in C's reference frame. Therefore, in this approach, each quantum clock constitutes a legitimate temporal (quantum) reference frame for which a notion of "evolution with respect to time $\tau$" is available. Importantly, this evolution is unitary, regardless of whether the clock is located far away or not from gravitating quantum systems, or whether the clock gets entangled with such gravitating degrees of freedom from the perspective of yet another time reference frame. This result is connected to the fact that, in a relational approach to physics, quantum superposition and entanglement become relative notions[25,33]. The fact that Eq. (10) maintains the same form when

we change from C's perspective to A's perspective means that the law for time evolution has a universal covariant form with respect to changes of time reference frames.

**Events with respect to gravitationally interacting clocks.** As discussed previously, each clock defines a time reference frame with respect to which a notion of time evolution can be defined. This is true even in the presence of gravitating quantum systems that lead to an indefinite metric background. What is the difference, in terms of the time localisation of events relative to an observer, between a situation with a definite metric background from one where such background is not fixed? Here we address this question by studying how the localisation of operationally defined events depends on the time reference frame. Consider the situation where A sets-up an event in her time reference frame by applying an operation on the system S at a particular time according to her clock. Suppose that the gravitational interaction between A and B is non-negligible. Suppose, for simplicity, that clock C is sufficiently far away from A and B so that we can neglect its gravitational interaction with them. This situation corresponds to the constraint

$$\left( \hat{H}_A + \hat{H}_B + \hat{H}_C + \lambda \hat{H}_A \hat{H}_B + \hat{f}(\hat{T}_A)(1 + \lambda \hat{H}_B) \right)|\Psi\rangle = 0, \quad (12)$$

where $\lambda = -G/(c^4 x_{AB})$. Here, $\hat{f}(\hat{T}_A)$ denotes a hermitian-operator-valued function modelling the interaction between the system of interest, S, and an ancilla a, recording the result of a measurement on S. The presence of the term $(1 + \lambda \hat{H}_B)$, multiplying $\hat{f}(\hat{T}_A)$, is due to the fact that making S and A's ancilla interact requires some energy, which will necessarily couple to B due to the universality of gravity.

Let us first analyse the history state in the time reference of C. As shown in Supplementary Note 2, this state is

$$|\Psi\rangle = \int d\tau_C \; |\tau_C\rangle_C \otimes \; e^{-i\tau_C(\hat{H}_A + \hat{H}_B + \lambda \hat{H}_A \hat{H}_B)}$$
$$\mathrm{T} e^{-i \int_0^{\tau_C} ds(1 + \lambda \hat{H}_B)\hat{f}(s(1 + \lambda \hat{H}_B) + \hat{T}_A)} |\psi_C(0)\rangle_{\bar{C}}. \quad (13)$$

Here, we have used the notation $\tau_C$ to emphasise that this is the time measured with respect to clock C, i.e., its proper time. Eq. (13) shows that, with respect to C, the event corresponding to the function $\hat{f}$ is not sharply defined in time, due to the presence of the operators in the argument of $\hat{f}$ in Eq. (13). Indeed, even if the initial state with respect to C is such that clock A is sharply defined at $t'_A = 0$, the presence of the operator $\hat{H}_B$ in the argument of $\hat{f}$ leads to an uncertainty in the time localisability of the event. More precisely, suppose that the initial state in C's frame is given by $|\psi_C(0)\rangle_{\bar{C}} = |t'_A = 0\rangle_A \otimes \int dt'_B \; \varphi_B(t'_B)|t'_B\rangle_B \otimes |\chi\rangle_R$ (R denotes the subsystem formed by S and a). Because B is a clock, it cannot be sharp in $\hat{H}_B$. Let $1/\sigma \neq 0$ be the characteristic width of $\tilde{\varphi}_B(\omega_B)$, the Fourier transform of $\varphi_B(t'_B)$. The history state of Eq. (13) is a coherent superposition, modulated by $\tilde{\varphi}_B(\omega_B)$, of terms containing $\hat{f}(s(1 + \lambda \omega_B))$ for different values of $\omega_B$. This case is mathematically similar to the case studied in the section "Non interacting clocks", with $\omega_B$ playing the role of $t'_B$. Because the trigger of the measurement depends on $\omega_B$, different values of $\omega_B$ correspond to different times at which the event happens (as described by C). Therefore, there will be an uncertainty of order $1/\sigma$ in the time localisation of the event (with respect to C). In fact, there is a type of uncertainty relation between the accuracy of clock B and the temporal localisability of events defined by clock A: The sharper clock B is localised in $\hat{T}_B$, the "fuzzier" the events defined by clock A appear from the point of view of C. This effect is depicted in Fig. 3.

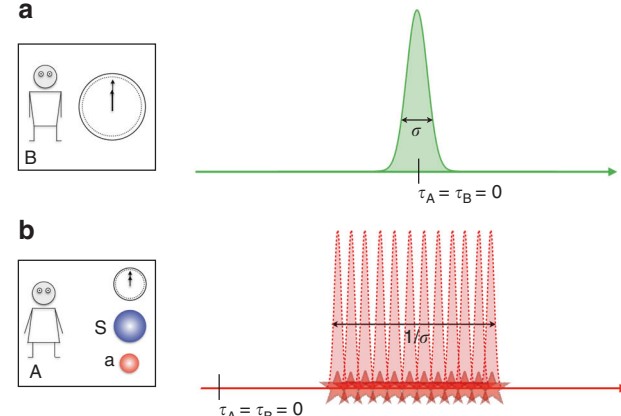

**Fig. 3 Gravitating quantum clocks from the point of view of C.** In a thought experiment, A (**b**) sets up an event, consisting in an interaction between S (blue ball) and a (red ball), when her clock shows a certain time $t_A^*$. A's clock is influenced by a gravitational field sourced by the energy of B's clock (**a**). The initial quantum state of B's clock (depicted by the green Gaussian) has a characteristic width $\sigma$, which specifies its accuracy (the smaller $\sigma$, the higher the accuracy). As a consequence, the energy of B's clock is not well defined—it has an uncertainty of $1/\sigma$. Therefore, the gravitational field sourced by B is not well defined either. As a consequence, the time dilation of clock A becomes uncertain from the point of view of C. This is shown by the "fuzzy" red wave packets representing A's clock state. By Eq. (13), this uncertainty leads to an indefiniteness in the localisation of A's event, as depicted by the "fuzzy" red stars on the bottom of the wave packets.

We now change to the time reference frame of A. First, we expand $|\psi_C(0)\rangle_{\bar{C}}$ in the time basis and plug it into Eq. (13). Then, we define the coordinate $\tau_A(\tau_C) := t'_A + \tau_C(1 + \lambda \omega_B)$ and make a change of variables to eliminate $\tau_C$ in favour of $\tau_A$. In "Methods (Remarks on coordinates for gravitating quantum systems)" we discuss the physical significance of coordinate transformations of this type, which can be understood as quantum coordinate transformations, and analyse them from a geometrical point of view. The history state in the reference frame of A reads (See Supplementary Note 2)

$$|\Psi\rangle = \int d\tau_A \; |\tau_A\rangle_A \otimes \mathrm{T} \; e^{-i \int_0^{\tau_A} ds \left( \frac{\hat{H}_B + \hat{H}_C}{1 + \lambda \hat{H}_B} + \hat{f}(s) \right)} |\psi_A(0)\rangle_{\bar{A}}, \quad (14)$$

where the relation between the initial state with respect to A, $|\psi_A(0)\rangle_{\bar{A}}$, and that with respect to C, $|\psi_C(0)\rangle_{\bar{C}}$, can be computed as shown in Methods (General method for changing reference frames). As in the non-interacting case, we assume that the support of the wave packets of clocks B and C do not overlap with the time defining the event.

As we can see from Eq. (14), the description of the history state in the time reference frame of A is such that the event is always sharp in $\tau_A$—$\hat{f}$ is a function of the c-number $s$ only. By contrast, Eq. (13) shows that the same event is not sharply localised from the point of view of C. This comparison shows that, when clocks interact gravitationally, the localisability of events in time is relative, and depends on the time reference frame which defines the events. Because it is not possible to "shield" gravity, this result suggests that the relativity of event localisability is a general feature of an operational definition of events in experiments with gravitating quantum systems. Moreover, if the interaction between the clocks were turned off, it would be possible to sharply define the time localisation of any event with respect to every time reference frame. The fact that we cannot do that,

practical reasons aside, is precisely because we have an indefinite metric background. This discussion suggests that, in this context, we can characterise a definite metric if we can find a time reference frame such that, with respect to it, every event (defined sharply with respect to some clock as we do in this paper) is sharply localised. This fact is deeply related to the result in ref. [27], developed in connection to process matrix framework[11]. In ref. [27] it is shown that the localisability of local operations in a causal reference frame is not absolute if and only if the corresponding process matrix is causally non-separable. For a discussion regarding time reference frames and local operations understood in the context of the process matrix framework see "Methods (Knob settings and external evolution)".

## Discussion

We have constructed a framework for temporal reference frames that associates a quantum clock to each time reference frame. Within this framework, we have shown that one can consistently define a meaningful notion of quantum operation, time evolution and localisation of events in time with respect to different quantum clocks. Importantly, this is true even in the case where these clocks interact gravitationally and the space-time metric is indefinite. We have studied how these notions change when we go from one time reference frame to another and found situations where the physical description of time evolution is covariant (form invariant) with respect to transformations between time reference frames, in line with ref. [25]. Our definition of an event allowed us to show that there is a time reference frame where the description of a given event assumes its familiar form in ordinary quantum mechanics (in terms of a unitary dilation of a quantum operation), even in cases where the metric field is indefinite. In the cases studied, the operations which are not localised with respect to the time reference frame acquire the form of a "quantum controlled unitary dilation", where the time parameter is replaced by an operator. It would be interesting to know exactly how our approach is related to Hardy's implementation of a "quantum equivalence principle"[10], where he proposes that the causal order can be made definite locally, around any given point.

We have used throughout the concept of a perfect clock. Although unrealistic, perfect clocks allowed us to explore the concepts of time evolution and time localisation of events without the difficulties associated with more realistic clock models. In experiments, these difficulties would clearly have to be taken into account, but the conceptual framework laid out here is independent of such problems. In fact, we consider our results as a basis of a more complete framework to describe phenomena in the absence of a well-defined space-time metric, especially suited for promising experimental realisations, like those proposed in refs. [34,35]. In this respect, the formulation of the process matrix framework in the form of ref. [27] leads one to speculate that such framework might not even need the explicit consideration of clocks.

An important effect occurs when quantum clocks interact gravitationally. In this case, we have shown that whether an event is localised in time or not depends on the time reference frame chosen. We have argued that the impossibility to find a time reference frame in which all events are local characterises, within our framework, a situation where the metric field is indefinite. This is relevant in the context of in refs. [26,27], where a connection between causal reference frames (or time delocalised subsystems) and pure process matrices[36] was established. It would be interesting to find out exactly how the frameworks of refs. [26,27] relate to our work, especially with the goal of understanding if a quantum violation of causal inequalities[11] can be achieved with gravitating quantum systems.

In this work, we have assumed that the spatial degrees of freedom of the gravitating quantum systems were in a semi-classical state, so that we could treat them classically. Although this simplification is useful, it would be important to extend our framework beyond this approximation. In a similar vein, it would also be interesting to generalise our framework to cases where terms of higher order in $1/c^2$ are included in the gravitational interaction.

Finally, on a more speculative note, our analysis of the gravitational quantum switch in "Methods (Gravitational quantum switch)" and its geometrical description in "Methods (Remarks on coordinates for gravitating quantum systems)" points to the fact that that, at least in the simplified case studied in this work, time reference frame transformations can be thought of as transformations on a space formed by a collection of manifolds, one per each classical solution arising from a different state of the gravitating quantum system. This observation suggests a new, convenient mathematical "arena", suitable for the analysis of physics with an indefinite metric. It is commonly agreed that, when quantum mechanics and general relativity are put together, the concept of space-time would have to yield to "something else". We hope that, using operational reasoning along the lines of the one presented in this paper, we can gain some insight into what this "something else" might be.

## Methods

**Review of the timeless approach to quantum mechanics.** Let us briefly review the "timeless" approach to quantum mechanics. This approach was proposed by Page and Wootters[20] and further developed by Giovanetti, Lloyd and Maccone[21]. Our review follows closely this latter version, which is particularly relevant to our work. A similar approach is developed by covariant quantum mechanics[23,24]. The basic starting point is the idea that time evolution "emerges" from relational degrees of freedom of quantum systems. Although classical dynamics admits a timeless formulation as well[22,37], here we focus on the quantum description, which is the one relevant to our work.

In the timeless formulation, one typically considers a total system composed of a clock, A, and the system of interest (the rest), R. The joint quantum system formed by A and R has a Hilbert space $\mathcal{H} = \mathcal{H}_A \otimes \mathcal{H}_R$, where $\mathcal{H}_A$ and $\mathcal{H}_R$, denote the Hilbert spaces of A and R, respectively. The time evolution of R with respect to A is encoded in the "history state", $|\Psi\rangle$, which contains all the information about the correlations between A and R, thus defining the dynamics of R with respect to the clock A. In the timeless approach, $|\Psi\rangle$ is subjected to a constraint, $\hat{C}$, acting on $\mathcal{H}$, as indicated in Eq. (2). The space of all states satisfying Eq. (2) is called the physical Hilbert space. Strictly speaking, it is not in general a subspace of the kinematical Hilbert space $\mathcal{H}$, because the inner product of the two spaces may differ. A convenient way for obtaining the history state, which we will use later on, is by "projecting" onto the physical Hilbert space, (see, e.g., ref. [38])

$$|\Psi\rangle = \int d\alpha \; e^{-i\alpha\hat{C}} |\varphi\rangle. \tag{15}$$

Eq. (15) gives a solution to Eq. (2) for any given $|\varphi\rangle \in \mathcal{H}$. $\hat{P} := \int d\alpha \; e^{-i\alpha\hat{C}}$ is not a projector in the mathematical sense, hence the quotation marks. In this paper, all integrals without specified limits are to be understood as integrals over $\mathbb{R}$.

In order to explain how time evolution emerges from Eq. (2), we analyse the simplest case, in which reading time $t_A$ on clock A corresponds to measuring the eigenvalue $t_A$ of an observable $\hat{T}_A$, acting on $\mathcal{H}_A$ in the same way the position operator acts in ordinary quantum mechanics. The joint system A + R obeys the constraint

$$(\hat{H}_A + \hat{H}_R)|\Psi\rangle = 0, \tag{16}$$

where the Hamiltonian of the clock, $\hat{H}_A$, is canonically conjugate to $\hat{T}_A$, $[\hat{T}_A, \hat{H}_A] = i$, and $\hat{H}_R$ acts on $\mathcal{H}_R$.

The ket $|\psi(t)\rangle := \langle t|_A \cdot |\Psi\rangle$ is called the reduced state, and has the physical interpretation of "the state of system R when clock A shows time $t$." It is easy to see that, by acting on Eq. (16) with $\langle t|_A\cdot$, the Schrödinger equation for $|\psi(t)\rangle$ with respect to time $t$ follows[21,28]. (Alternatively, we can find $|\Psi\rangle$ directly by means of Eq. (15) and show that the reduced state undergoes a unitary time evolution.) It is in this sense that time evolution is recovered from the "timeless" condition given by Eq. (2). Eq. (16) describes a situation in which time evolution is specified by a "perfect clock". By definition, a perfect clock is a system satisfying the following: (i) It is associated with an operator $\hat{T}_A$, acting on an infinite dimensional Hilbert space $\mathcal{H}_A$. (ii) Its Hamiltonian, $\hat{H}_A$, is canonically conjugate to $\hat{T}_A$. Although not realistic (its Hamiltonian is unbounded from below), a perfect clock is a convenient

idealisation for the purposes of this paper. It can be considered (although it need not be) as an approximation of an $n$-level finite-dimensional system in the limit where $n$ is very large[39]. In this work we use the concept of a perfect clock as a convenient abstraction, which will help us to capture the essential features of the time-ordering of events and time evolution in the presence of gravitating quantum systems. By using this abstraction, we can circumvent the difficulties associated to defining the time localisation of events with realistic clocks[40], and focus on the status of the notion of an event and its "space-time" localisation in the absence of a definite gravitational field.

Describing measurements performed at multiple times in the timeless framework is a subtle issue. Naively applying the projection postulate of quantum mechanics to $|\Psi\rangle$ would, in general, produce a post-measurement state that violates Eq. (2). However, as noted in refs. [21,24], a consistent way of describing multiple measurements in a timeless framework is to "purify" them. This means dividing the subsystem R into the system of interest, S, and a set of ancillary systems $\{a_i\}$. The ancilla $a_i$ acts as measurement device that records the information about the i-th measurement performed on S. The purification consists in modelling the measurement on S by explicitly including an interaction Hamiltonian between S and the ancillary systems $a_i$ in the constraint $\hat{C}$. After the interaction, the result of the measurement is revealed by independently measuring in the state of the ancillas. For example, a simple model of a multiple-time measurement, happening at times $t_1$ and $t_2$, with $t_1 < t_2$, corresponds to the constraint equation

$$(\hat{H}_A + \hat{H}_S + \delta(\hat{T}_A - t_1)\hat{K}^{(1)} + \delta(\hat{T}_A - t_2)\hat{K}^{(2)})|\Psi\rangle = 0. \qquad (17)$$

Here, $\hat{H}_S$ is the free Hamiltonian of the system of interest, S, $\delta$ denotes the Dirac delta distribution, and $\hat{K}^{(i)}$, for I = 1, 2, is an operator acting on $\mathcal{H}_S \otimes \mathcal{H}_{a_i}$, where $\mathcal{H}_{a_i}$ is the Hilbert space associated to the i-th ancilla. The history state describes measurement interactions between the system S and the ancillas $a_i$. These interactions are sharply localised at times $t_1$ and $t_2$. At times where the interactions are turned off, S evolves freely under the Hamiltonian $\mathcal{H}_S$[21].

The probabilities for the results of measurements are given by projecting the history state only once. For example, the probability for obtaining a result $a_1$, corresponding to the measurement performed at $t_1$, and a result $a_2$, corresponding to the measurement performed at $t_2$, given that clock A shows a time $t > t_2$, is[21]:

$$p(a_1, a_2|t > t_2) = |\langle t|_A \otimes \langle a_1| \otimes \langle a_2| \cdot |\Psi\rangle|^2. \qquad (18)$$

By assumption, the state $|\psi(t)\rangle := \langle t|\Psi\rangle$ is normalised to one. Note that there is nothing special about the choice of time $t > t_2$. If one wishes, one can also compute the probabilities for the outcome of the first measurement only, by projecting at a time $t_1 < t < t_2$. Moreover, if one is interested in measurements at times $t'_1$ and $t'_2$, different from $t_1$ and $t_2$, one can simply replace $t_1$ and $t_2$ by $t'_1$ and $t'_2$ in Eq. (17), and compute the desired probabilities by means of Eq. (18).

**Changing time reference frames**. Here we develop a method for computing the history state for one observer, say A, given the history state with respect to another observer, say C. In Supplementary Note 2, we find all the history states studied in this paper by group averaging—using Eq. (15). This leads to the "perspective neutral" representation of $|\Psi\rangle$, which we use to "jump" into the time reference frame of A. The perspective neutral representation offers an approach for changing from the time reference frame of A to that of B. Moreover, it underscores the interesting fact that "jumping" into a (classical or quantum) reference frame amounts to fixing the redundant degrees of freedom imposed by the constraint $\hat{C}$ (see refs. [17,41–45]). It may happen that one wishes to change from the time reference frame of A to that of B without explicitly computing the perspective neutral representation of $|\Psi\rangle$, as we do in Supplementary Note 2. Here, we find explicit equations that allow us to do so. Note that the two ways of computing the history state give the same result. This fact was pointed out recently in ref. [46], which builds upon the method for changing relational quantum clocks via a sequence of "trivialisation maps" developed in ref. [17].

As noted in the main text, the timeless approach is based on a constraint given by Eq. (2). We can obtain $|\Psi\rangle$, the solution to Eq. (2), by "projecting" an arbitrary state $|\varphi\rangle$ onto the space of solutions of Eq. (2), as done in Eq. (15). That is

$$|\Psi\rangle = \hat{P}|\varphi\rangle, \qquad (19)$$

where $\hat{P} = \int d\alpha\, exp(-i\alpha\hat{C})$. Strictly speaking, $\hat{P}$ is not a projector: it maps the kinematical Hilbert space $\mathcal{H}$ onto a physical Hilbert space formed by solutions of Eq. (2). As mentioned before, the physical Hilbert space is, in general, not isomorphic to the kinematical one, because the inner products of each space might be different to each other.

We can extract operationally meaningful physical predictions from $|\Psi\rangle$ by expressing it in a specific time reference frame (say A or C, for concreteness). As we have seen in the main text, there are interesting cases where the time evolution with respect to either A or C is unitary. In these cases, we have

$$\langle t|_I\, \hat{P}\, |t'\rangle_I = \hat{U}_{\bar{I}}(t - t'), \qquad (20)$$

where I can be either A or C (the time reference frames analysed in the main text), and $\hat{U}_{\bar{I}}(t - t')$ is the evolution operator in the time reference frame of I. This operator is unitary and acts on all degrees of freedom except for I's. Like all

evolution operators, it satisfies the composition property: $\hat{U}_{\bar{I}}(t - t') = \hat{U}_{\bar{I}}(t)\hat{U}_{\bar{I}}(-t') = \hat{U}_{\bar{I}}(t)\hat{U}_{\bar{I}}^\dagger(t')$. We will focus on the cases where Eq. (20) holds.

We can express $|\Psi\rangle$ it in the time reference frame of C by plugging $|\varphi\rangle = \int dt'_C |t'_C\rangle_C \otimes |\psi_C(t'_C)\rangle_{\bar{C}}$ into Eq. (19), and then inserting a resolution of the identity, $\mathbb{1} = \int dt_C |t_C\rangle\langle t_C|_C$ on the right hand side of $\hat{P}$. Using Eq. (20), we obtain

$$|\Psi\rangle = \int dt_C\, |t_C\rangle \otimes \hat{U}_{\bar{C}}(t_C)|\psi_C(0)\rangle_{\bar{C}}, \qquad (21)$$

where

$$|\psi_C(0)\rangle_{\bar{C}} = \int dt'_C\, \hat{U}_{\bar{C}}^\dagger(t'_C)|\psi_C(t'_C)\rangle_{\bar{C}}. \qquad (22)$$

Note that we have used the composition property of $\hat{U}_{\bar{C}}$ in order to write down Eqs. (21) and (22). Note also that, when introducing the resolution of identity, we have assumed that, in the physical Hilbert space, the integration measure in the position (or rather time) representation is given by $dt_C$. This is the case in all instances considered in this work and we will assume it in the following. In more general cases, one simply has to consider the specific form of the integration measure when inserting resolutions of identity in the time representation. In general, because the state $|\psi_I(0)\rangle_{\bar{I}}$ evolves unitarily with respect to an arbitrary clock I, its normalisation is preserved in time with respect to this arbitrary frame.

We can obtain $|\Psi\rangle$ in the time reference frame of C in different ways. For example, we can obtain first $|\Psi\rangle$ in a "perspective neutral" representation, by computing explicitly the integral with respect to $\alpha$ in Eq. (19), and afterwards, by using the methods of refs. [17,41,45], we can "jump" into the reference frame of C. Alternatively, we can carry out explicitly the computations leading from Eq. (19) to Eq. (21), or we can act with $\langle t_C|_C \cdot$ on Eq. (2) and solve the resulting differential equation. In any case, suppose we are given $|\Psi\rangle$ in the time reference frame of C, that is, in the form of Eq. (21), and we would like to change the representation to the time reference frame of A. We can do so directly by noting that

$$\hat{P} = \int dt_C\, dt'_C\, |t_C\rangle\langle t'_C|_C \otimes \hat{U}_{\bar{C}}(t_C - t'_C). \qquad (23)$$

Therefore, by Eq. (20), we have

$$\hat{U}_{\bar{A}}(t_A - t'_A) = \int dt_C\, dt'_C\, |t_C\rangle\langle t'_C|_C \otimes \langle t_A|_A \hat{U}_{\bar{C}}(t_C - t'_C)|t'_A\rangle_A. \qquad (24)$$

Note that Eq. (24) allows us to obtain the (unitary) evolution operator in the time reference frame of A directly from the evolution operator in the reference frame of C. All we need to obtain $|\Psi\rangle$ in the time reference frame of A is an equation for the initial state with respect to A, $|\psi_A(0)\rangle_{\bar{A}}$, in terms of that of C. But this is easy because, by definition, $|\psi_A(0)\rangle_{\bar{A}} = \langle t_A = 0|_A \cdot |\Psi\rangle$. Then, by Eq. (21), we have

$$|\psi_A(0)\rangle_{\bar{A}} = \int dt_C\, |t_C\rangle_C \otimes \langle t_A = 0|_A \hat{U}_{\bar{C}}(t_C)|\psi_C(0)\rangle_{\bar{C}}. \qquad (25)$$

Equations (24) and (25) are all we need to change the time reference frame representation of $|\Psi\rangle$. Note that, in principle, $\hat{S}_{AC} = \int dt_C\, |t_C\rangle_C \otimes \langle t_A = 0|_A \hat{U}_{\bar{C}}(t_C)$, which transforms between the initial states of C and A in Eq. (25), need not be unitary. In the simplest case, where $\hat{C} = \hat{H}_A + \hat{H}_C + \hat{H}_S$, we have

$$\hat{S}_{AC} = \mathcal{P}_{AC} e^{i\hat{T}_A \hat{H}_S}, \qquad (26)$$

where $\mathcal{P}_{AC} := \int dt_C\, |t_C\rangle_C\langle -t_C|_A$ is the parity-swap operator between C and A. Note that this transformation matches exactly the transformation introduced in ref. [25], for transforming between two spatial quantum reference frames.

Finally, if we wish to change from the reduced state at time $t_C$ in C's frame, $|\psi_C(t_C)\rangle_{\bar{C}} = \langle t_C|_C \cdot |\Psi\rangle$, to the reduced state at time $t_A$ in A's frame, $|\psi_A(t_A)\rangle_{\bar{A}} = \langle t_A|_C \cdot |\Psi\rangle$, we simply have to put together Eqs. (21) and (25):

$$|\psi_A(t_A)\rangle_{\bar{A}} = \hat{U}_{\bar{A}}(t_A)\hat{S}_{AC}\hat{U}_{\bar{A}}^\dagger(t_C)|\psi_C(t_C)\rangle_{\bar{C}}. \qquad (27)$$

In general, this transformation is not unitary, because $\hat{S}_{AC}$ is not always unitary.

**Operational meaning of the framework**. Here we explain in detail the operational meaning of our framework, described briefly in Reference frames for events and time evolution. In Fig. 1, two observers, A and B, perform experiments on a quantum system S. Each of them possess a clock and an ancilla, labeled a for A and b for B.

The experiment has two stages, "preparation" and "detection". In the preparation stage, A and B prepare the states of their clocks, ancillas and systems. They can, for example, choose to set their clocks at $t = 0$, and set S in a particular state. S can be a composite system with subsystems accessible to A and B, and the observers can prepare S in an entangled state in the partition defined by these subsystems. In the detection stage of the experiment, A and B analyse the measurement results by looking at the outcome of their clocks and ancillas. For example, they might be interested in checking whether the ancillas are in a certain state at a given time, as

defined by one of the clocks. Alternatively, they might project one of the clocks, say A, in a specific basis, which need not be the time basis $\{|t_A\rangle_A\}$.

The clocks and ancillas allow A and B to produce events localised in time, as defined by the reading of their own clocks. For the purposes of this work, an event consists in any quantum operation performed on a quantum system, conditioned on one of the clocks being in a state which is sharply localised around a specific time. For example, A can perform an experiment such that, when her clock shows time $t_A^*$, an entangling unitary $\hat{U}$ between S and a occurs, recording information of S in a. If the outcome of a reveals information about S, the event corresponds to a measurement of S by means of a. An event is not necessarily a measurement. For example, the application of a unitary operation on S alone is also an event. However, in order for this event to be defined operationally, the information that such unitary took place has to be recorded in a physical system (a counter) to which A and B have access in the detection stage. For example, the counter can be a two-level system which shows 1 if the unitary has been applied and shows 0 otherwise. Our definition of "event" is adapted to the features of general relativity, where diffeomorphism invariance means that points in space-time have no physical meaning on their own, and events have to be defined via coincidences of physical fields[8,47].

In this work we insist on such an operational definition of events even in the case where no space-time background is assumed. That is, we consider the general case where A and B do not assume that the experiments take place in a specific, well-defined causal structure. This could be because of practical reasons—they might have only a probabilistic knowledge about the space-time geometry where the experiments take place—or because of fundamental reasons—they might do experiments involving gravitating quantum systems, which can lead to situations with indefinite causal structure[12]. In any case, A and B know that they can make quantum mechanical predictions by using the timeless approach to quantum mechanics, and are able to verify which is the constraint equation and the history state corresponding to their experiments. By keeping track of the different events in the experiment, and comparing the times at which these events occur with respect to their own clocks, A and B can assign a time coordinate to each event. In this way, they can define a "mapping" of the set of events into "space-time", as shown in Fig. 1.

After repeating the experiment several times, A and B will be able to track the time, relative to their respective clocks, at which the events occur. For example, B can find that the aforementioned unitary $\hat{U}$ takes place sharply when his clock shows the time $t_B^*$, which in general differs from $t_A^*$. In the case where the event corresponds to a measurement, B can identify such event in terms of a statement like "at time $t_B^*$ a measurement was performed on S yielding the result 'up'".

**Gravitational quantum switch.** In this section we analyse the gravitational quantum switch[12] from the perspective of time reference frames. The gravitational switch is a thought-experiment where a gravitating body put in a quantum superposition of positions leads to an indefinite causal order of events. The experimental set up is as follows (see Fig. 4). Two parties, A and B, perform operations on a quantum system S (one operation each party). Each operation happens when the corresponding local clock shows time $t^*$. Apart from A and B's clocks, there is a mass M, prepared in a superposition of two different position

eigenstates, labeled by L (left) and R (right), as described in the reference frame of C. Note that here R stands for "right", in contrast to the rest of the paper, where it stands for "rest". We assume that the superposition has the same weight for L and R. Each amplitude L and R leads to two different metric fields, and therefore to two different space-times. It is useful to keep track of the two different space-times explicitly, by denoting them $\mathcal{M}_L^{(C)}$ (for the mass on the left) and $\mathcal{M}_R^{(C)}$ (for the mass on the right). The experimental set up is such that, for the amplitude corresponding to L (R), M is closer to A (B). Because of gravitational time dilation, in the spacetime $\mathcal{M}_R^{(C)}$ ($\mathcal{M}_L^{(C)}$), the event where A's clock shows $t^*$ and A acts on S is in the past (future) light cone of the event where B's clock shows $t^*$ and B acts on S. Thus, the gravitational switch leads to a superposition of causal orders. For simplicity, we assume that the distance between M and A in the configuration given by L is the same as the distance between M and B in the configuration given by R, and the distance between A and B is constant.

We now use the framework of time reference frames to analyse and give a geometric description of the gravitational switch. We proceed by writing down the constraint describing the experiment and then comparing the history state in the reference frame of C to that in the reference frame of A (the case of B is analogous to that of A). The Hilbert space of the whole system is $\mathcal{H} = \mathcal{H}_A \otimes \mathcal{H}_B \otimes \mathcal{H}_C \otimes \mathcal{H}_S \otimes \mathcal{H}_a \otimes \mathcal{H}_b \otimes \mathcal{H}_M$, where A, B, and C denote the clocks of the three different time reference frames, S is the system on which A and B perform operations, and a and b are the ancillas that record the measurement outcomes of A and B, respectively. M denotes the relative degrees of freedom between the massive system and the clocks. For simplicity, we focus on the subspace generated by relative position eigenstates, labeled by Q = L, R, denoting "mass close to A" and "mass close to B", respectively. In the framework of time reference frames, this experiment is described by the constraint equation

$$\left(\sum_I \hat{H}_I(1+\hat{\Phi}_I) + \sum_I \hat{f}_I(\hat{T}_I)(1+\hat{\Phi}_I)\right)|\Psi\rangle = 0, \qquad (28)$$

where I takes the values A, B and C. We consider sharply localised measurements, $\hat{f}_A(\hat{T}_A) = \delta(\hat{T}_A - t^*)\hat{K}_{Sa}^{(A)}$, with an analogous definition for B. By assumption, $\hat{f}_C = 0$. Here, $\hat{\Phi}_I = -GM/(c^2\hat{x}_{IM})$ denotes the gravitational potential, M is the mass of the system put in a superposition of locations, and $\hat{x}_{IM}$ is the relative distance operator between the mass and clock I. By definition, the operator $\hat{\Phi}_I$ acts on $\mathcal{H}_M$ as $\hat{\Phi}_I|Q\rangle_M = \Phi_I^Q|Q\rangle_M$. Note that $\mathcal{H}_M$ is generated by the eigenstates of the operators $\hat{x}_{IM}$, for I = A, B, C. The states $|Q\rangle_M$ are therefore eigenstates of each $\hat{x}_{IM}$, for I = A, B, C. For simplicity, we assume that C is located equidistant from both locations of the mass, so that $\Phi_C^L = \Phi_C^R =: \Phi_C$.

Let us now find the history state, $|\Psi\rangle$, of Eq. (28) in the reference frames of A and C. In this case it is instructive to find first $|\Psi\rangle$ in a "perspective neutral" representation, and then use this representation to move to the time reference frames of A and C. By inserting the constraint $\hat{C}$ of Eq. (28) into Eq. (15) we find (see Supplementary Note 1)

$$|\Psi\rangle = \int d\alpha \, e^{-i\sum_{I=A,B,C} \hat{H}_I(1+\hat{\Phi}_I)} \, \mathrm{T} \, e^{-i\sum_{I=A,B} \int_0^\alpha ds(1+\hat{\Phi}_I)\hat{f}(s(1+\hat{\Phi}_I)+\hat{T}_I)} |\varphi\rangle. \qquad (29)$$

The description of the quantum switch given in Eq. (29) can be roughly interpreted as the quantum state seen by a distant observer who uses the coordinate $\alpha$ as parameter time. (More precisely, if we introduce yet another party, which is sufficiently far away that it effectively does not interact with the rest, the description of the quantum switch would be given by Eq. (29) plus the additional degrees of freedom of the non-interacting party). However, this interpretation is not physically rigorous, because the constraint (28) implies that there is no external time parameter according to which the systems evolve. For this reason, we write down the history state from the perspective of C. In order to do so, we expand $|\varphi\rangle$ in the $|t'_A, t'_B, t'_C\rangle_{ABC} \otimes |Q\rangle_M \otimes |\chi\rangle_{Sab}$ basis and follow the steps outlined in Supplementary Note 2. The result is

$$|\Psi\rangle = \int d\tau_C \, |\tau_C\rangle_C \otimes \mathrm{T} \, e^{-i\sum_I \int_0^{\tau_C} ds\hat{\Delta}(I,C)(\hat{H}_I+\hat{f}_I(s\hat{\Delta}(I,C)+\hat{T}_I))} |\psi_C(0)\rangle_{\bar{C}}, \qquad (30)$$

where the sum is over I = A, B. We have defined $\hat{\Delta}(I,J) := (1+\hat{\Phi}_I)/(1+\hat{\Phi}_J)$, for I, J = A, B, C. Note that $\hat{\Delta}(I,J)$ is the operator version of the usual redshift factor, formed by the ratio of the proper time of clock I to that of clock J. We denote the eigenvalues of $\hat{\Delta}(I,J)$ by $\Delta^Q(I,J)$, where $\hat{\Delta}(I,J)|Q\rangle_M = \Delta^Q(I,J)|Q\rangle_M$. In this way, $\Delta^Q(I,J)$ is the ratio of the proper time of clock I to that of clock J in the mass configuration Q. As in previous cases, we assume that the initial state, $|\psi_C(0)\rangle_{\bar{C}}$, has both clocks A and B sharply localised around $t_A = t_B = 0$. The reason for this assumption is simply that we are more interested in the effects due to the gravitating quantum system M than in those due to the "fuzziness" of the initial state. The relation between $|\varphi\rangle$ and the initial state $|\psi_C(0)\rangle_{\bar{C}}$ is given in Supplementary Note 2.

The fact that $\hat{\Delta}(I,C)$ is an operator acting on $\mathcal{H}_M$ has important consequences for the localisation of events in the reference frame of C. Indeed, the time ordering operator in Eq. (30) enforces that the operations of A and B are applied in different orders for the amplitudes corresponding to Q = L and Q = R. Specifically, for Q = L (Q = R) we have $\Delta^L(A,C) < \Delta^L(B,C)$ ($\Delta^R(A,C) > \Delta^R(B,C)$), so that A's operation

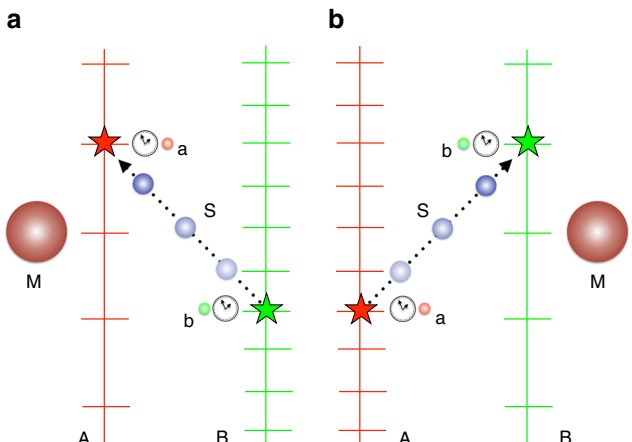

**Fig. 4 Experimental set up of the gravitational switch.** The mass M can be either close to A (**a**) or to B (**b**). A (B) applies an operation on the system S by means of an ancilla a (b). A's operation is in the causal future (past) of B's operation when the mass is on the left (right). This fact is depicted by the system traveling from B to A (**a**) and from A to B (**b**). In the quantum switch experiment, the mass is put in a superposition between being close to A and close to B, leading to a superposition of causal orders.

is applied after (before) B's. Note that, for simplicity, we are working in an approximation where a given event is in the causal past of another one if the time associated to the former is earlier than the time associated to the latter. In Supplementary Note 2 we comment on a possible way of taking a more complete approach, in terms of the light cones emanating from each event. In the present case, our simplification just means that, by assumption, the event triggered by clock A in the configuration $Q = L$ ($Q = R$) is in the future light cone (past light cone) of the event triggered by clock B.

We now compute the reduced state of system $\bar{C}$, $|\psi(\tau_f)\rangle_{\bar{C}} := \langle \tau_f|_C \cdot |\Psi\rangle$, at a time $\tau_f > t^*/\Delta^R(A, C) = t^*/\Delta^L(B, C)$, when both operations have already occurred in C's time reference frame. From Eq. (30), we have

$$|\psi(\tau_f)\rangle_{\bar{C}} = \frac{1}{\sqrt{2}} \int \frac{dt'_A dt'_B dt'_C}{1 + \Phi_C} \, \varphi(t'_A, t'_B, t'_C) (|\phi_L\rangle_{AB} \otimes |L\rangle_M \otimes \hat{U}^A_{aS}(t^*/\Delta^L(A, C)) \hat{U}^B_{bS}(t^*/\Delta^L(B, C)) + |\phi_R\rangle_{AB} \otimes |R\rangle_M \otimes \hat{U}^B_{bS}(t^*/\Delta^R(B, C)) \hat{U}^A_{aS}(t^*/\Delta^R(A, C))) |\chi\rangle_{abS},$$

(31)

where $\left|\phi_Q\right\rangle_{AB} = \bigotimes_{I=A,B} |t'_I + \Delta^Q(I, C)(\tau_f - t'_C)\rangle_I$, for $Q = L, R$. Here, the operator $\hat{U}^A_{aS} = exp(-i\hat{K}^A_{aS})$ represents the interaction by which the outcome of the event "the system S is measured when clock A shows $t = t^*$" is recorded in the ancilla a. Although this operator is time independent, we have written in both amplitudes the time, in C's reference frame, at which it was applied. This is in order to emphasise the time labelling of events in this reference frame. Similar remarks hold for the case of B. The state of Eq. (31) is nothing but the familiar description of the quantum switch[48] after the local operations of A and B have been performed and before the control system, formed in this case by the mass and the clocks of A and B, is recombined.

The description of the experiment according to the time reference frame of C is depicted in Fig. 5, where we have used two different manifolds to depict the two different space-times $\mathcal{M}_L^{(C)}$ and $\mathcal{M}_R^{(C)}$. As the figure shows, no event happens sharply localised in time according to C. Rather, the event corresponding to A occurs in a superposition in time between $\tau_C = t^*/\Delta^R(A, C)$ (early) and $\tau_C = t^*/\Delta^L(A, C)$ (late), with a similar statement applying to the case of B. Note that, according to our framework and its physical interpretation, the claim that the events in the switch experiment "involve 4 spacetime points" is a frame dependent statement—valid for C in this case. This is the natural picture of the gravitational switch that emerges if one imagines the experiment as seen by an observer far-away from the mass. However, note that, in our analysis, no assumptions regarding the distance from C to M were needed in order to obtain this description. Of course, we work with states on $\mathcal{H}_M$ such that no divergences in the redshift factors $\Delta^Q(I, J)$ occur. This is ensured as long as $x_{IM} > GM/c^2$ (the order of magnitude of the Schwarzschild radius corresponding to a mass of magnitude M.)

Let us now turn to the description of the experiment in the time reference frame of A (the case for B being completely analogous). From Eq. (29), the quantum controlled change of coordinates $\alpha \longrightarrow \tau_A := t_A + \alpha(1 + \phi_A^Q)$ gives the history

state in A-representation

$$|\Psi\rangle = \int d\tau_A \, |\tau_A\rangle_A \otimes \, T \, e^{-i \int_0^{\tau_A} ds (\hat{f}_A(s) + \sum_I \hat{\Delta}(I, A)(\hat{H}_I + \hat{f}_I(s\hat{\Delta}(I, A) + \hat{T}_I)))} |\psi_A(0)\rangle_{\bar{A}},$$

(32)

where the index I in the exponent takes values I = B, C. The relation between $|\psi_A(0)\rangle_{\bar{A}}$ and the state $|\varphi\rangle$ is found explicitly in Supplementary Note 2.

Note that the argument of $\hat{f}_A$ in Eq. (32) does not depend on any redshift factor, classical or quantum. This means that, according to A, her operation is always localised in time. This fact can be seen clearly when writing down the conditional state $|\psi(\tau_f)\rangle_{\bar{A}} := \langle \tau_f|_A \cdot |\Psi\rangle$ for a time $\tau_f > t^*/\Delta^R(B, A)$:

$$|\psi(\tau_f)\rangle_{\bar{A}} = \frac{1}{\sqrt{2}} \int \frac{dt'_A dt'_B dt'_C}{1 + \hat{\Phi}_A} \, \varphi(t'_A, t'_B, t'_C)|\phi\rangle_C \otimes (|\phi_L\rangle_B \otimes |L\rangle_M \otimes \, \hat{U}^A_{aS}(t^*) \hat{U}^B_{bS}(t^*/\Delta^L(B, A)) + |\phi_R\rangle_B \otimes |R\rangle_M \otimes \, \hat{U}^B_{bS}(t^*/\Delta^R(B, A)) \hat{U}^A_{aS}(t^*)) |\chi\rangle_{abS}.$$

(33)

For simplicity of notation, we have written $\tau_f$ to refer to the "final" time when analysing the experiment both from A's (Eq. (33)) and C's (Eq. (31)) perspective. However, these two times need not be the same. Note that here $|\phi\rangle_C = |t'_C + \Delta(C, A)(\tau_f - t'_A)\rangle_C$ factors out from the mass degrees of freedom due to the assumption $\Phi_C^L = \Phi_C^R$. On the other hand, the state of clock B is entangled with the mass. The important point is that, in Eq. (33), the operation $\hat{U}^A_{aS}$ depends only on $t^*$ in both amplitudes $Q = L, R$. As noted before, this means that the operation happens sharply at $t^*$ in the reference frame of A, independent of where the mass is. However, the operation $\hat{U}^B_{bS}$ occurs before (after) $\hat{U}^A_{aS}$ for the configuration $Q = L$ ($Q = R$). Therefore, for A, events in the vicinity of her clock are always well defined in time, whereas events outside this vicinity are "spread out" in her time[27]. As noted in the previous Section, this is a signature of an indefinite metric field. An indefinite metric field can lead to an indefinite causal order of events if, like in this case, the events are suitably chosen.

The situation described by Eqs. (32) and (33) is depicted in Fig. 6, where a geometric description of the experiment from the point of view of A is given. As noted before, A's operation takes place at time $t^*$ in both amplitudes.

**Remarks on coordinates for gravitating quantum systems.** Let us make a few (speculative) remarks regarding our treatment of coordinates:

1.  (Reference frames associated with a set of manifolds.) The geometric picture of the gravitational switch shown in Figs. 5 and 6 suggests that a time reference frame should not be considered as "attached" to a single space-time manifold. Rather, a time reference frame should be defined

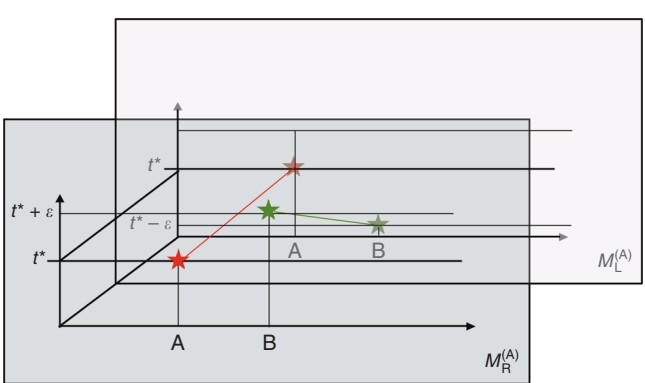

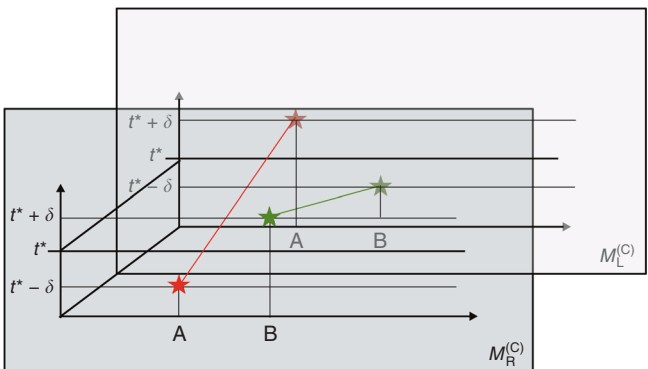

**Fig. 5 Space-time diagram for gravitational switch thought experiment as described in the time reference frame of C.** The event in which A (B) acts on S is depicted by a red (green) star. These events are delocalised in time from C's point of view. In the spacetime $\mathcal{M}_R^{(C)}$ ($\mathcal{M}_L^{(C)}$), the mass M is on the right (left), implying that A acts before (after) B. (For clarity, the worldline of M is not shown.) In $\mathcal{M}_R^{(C)}$ ($\mathcal{M}_L^{(C)}$), A's action happens at time $t^*/\Delta^R$ (A, C) $\approx t^* - \delta$ (resp. $t^*/\Delta^L$(A, C) $\approx t^* + \delta$) and B's action happens at time $t^*/\Delta^R$ (B, C) $\approx t^* + \delta$ (resp. $t^*/\Delta^L$(B, C) $\approx t^* - \delta$), for $\delta = (\Phi_C^L - \Phi_A^L)/t^*$. The dashed red (green) line joining A's (B's) event in both spacetimes is drawn to emphasise that this is the same event, even though we have used two red (green) stars, one per spacetime, to depict it.

**Fig. 6 The gravitational switch thought experiment as described in the time reference frame of A (local observer).** The event in which A acts on S is depicted by a red star. This event is perfectly localised in time from A's point of view. However, as in the previous case, in the spacetime $\mathcal{M}_R^{(A)}$ ($\mathcal{M}_L^{(A)}$), the mass M is on the right (left), implying that A acts before (after) B. The causal order is conserved, for each spacetime amplitude, after the change of perspective. In both $\mathcal{M}_R^{(A)}$ and $\mathcal{M}_L^{(A)}$, A's action happens at time $t^*$, whereas B acts at $t^*/\Delta^R$(B, A) $\approx t^* + \epsilon$ (resp. $t^*/\Delta^L$(B, A) $\approx t^* - \epsilon$) in $\mathcal{M}_R^{(A)}$ ($\mathcal{M}_L^{(A)}$), for $\epsilon = (\Phi_B^L - \Phi_A^L)/t^*$. Note that the dashed red line joining A's event in both spacetimes is now parallel to the dashed black line joining $t^*$ in both spacetimes. This means that we are in the time reference frame where the event happens at a precise, sharp time. In contrast, the dashed green line joining B's event in both spacetimes is not parallel to the black dashed one, showing that the time localisation of B's event is not sharp in A's time reference frame.

"transversally", by "piercing" through the different manifolds. This resonates with Hardy's idea[10] of quantum coordinate systems as an identification of points along different manifolds (this is depicted by the black, dashed lines in Figs. 5 and 6, which identify the time $t^*$ in the two different space-times).

2. (Hypersurfaces of constant $t$.) The previous remark suggests that, when writing a history state in the form $|\Psi\rangle = \int dt\, |t\rangle \otimes |\psi(t)\rangle$, the ket $|\psi(t)\rangle$ should not be interpreted as a state living on (the Hilbert space corresponding to a spatial slice of) a single spacetime, but rather on the hypersurface of "constant $t$" that "pierces" through the set of manifolds. The state on a single manifold is obtained by restricting $|\psi(t)\rangle$ to a single spacetime. (For the gravitational switch, this restriction would correspond to, roughly speaking, "projecting" into the subspace corresponding to the spacetime where the mass is, say, on the right.)

3. (Signature of an indefinite metric and robustness of the global causal structure) We emphasise that, in experiments involving gravitating quantum systems, the fact that events are delocalised in time with respect to some observers is a signature of an indefinite metric field—ignoring uncertainties due to "fuzzy" clock pointers and measurement interactions extended in time. More specifically, if the metric field is indefinite, one cannot, in general, change to a time reference frame where all events are localised in time. For example, in the gravitational switch, localising the event of A means that the event of B is uncertain in time. However, time reference frame transformations cannot change the global causal structure of the events in an experiment (either form A's or C's perspective, there is an amplitude where A's event is in the past of B's and an amplitude where A's event is in the future of B's). In other words, the localisability of a single event in time is a relative concept, whereas the global causal structure of events is an absolute one. This is reminiscent of what happens in the case of process matrices, where continuous and reversible transformations do not modify the global causal order of a process matrix[49].

4. (Geometric "arena" for superpositions of semiclassical states.) We point out that the above geometric picture, with multiple manifolds identified by time reference frames "piercing" through them, suggests a new suitable geometric "arena" in which phenomena with "superpositions of spacetimes" can be studied mathematically, at least in the case where the gravitating quantum systems are in a quantum superposition of semiclassical states. It would be interesting to investigate if an extension of this geometric picture can be useful beyond this case as well.

5. (Quantum coordinates.) It is interesting to note that we can understand the application of operations in Eq. (13) as being done with respect to a "quantum coordinate". Specifically, following the definition of an integral with respect to an operator given in non commutative analysis[32], we note that the integral part of the exponent in Eq. (13) can be written as $\int_0^{\tau_C} ds(1 + \lambda \hat{H}_B)\, \hat{f}(s(1 + \lambda \hat{H}_B) + \hat{T}_A) = \int_0^{\tau_C} d\hat{\tau}_A\, \hat{f}(\hat{\tau}_A)$, where $\hat{\tau}_A = s(1 + \lambda \hat{H}_B) + \hat{T}_A$ is the "quantum coordinate" upon which the integral is defined. This is the integral version of the "derivative" with respect to a quantum coordinate expressed by Eq. (9).

6. (Quantum-controlled change of coordinates.) Now we comment on the change of coordinates that eliminates $\tau_C$ in favour of $\tau_A$, leading from Eq. (13) to Eq. (14). Although mathematically very simple, this change of variables has an important physical interpretation. Because it associates a different $\tau_A$ to different amplitudes of the state of clock B, this change of coordinates is quantum-controlled by the state of B. Note that B is a source of the gravitational field, and is in a state which contains different amplitudes corresponding to different energies. Importantly, each of these energies corresponds to a different metric field, and therefore, to a different space-time. Then, the change of coordinates that goes from C's to A's frame associates different values of $\tau_A$ to different metric fields, each of which corresponds to a different amplitude in B's state. In this sense, it is more general than the usual coordinate transformations in general relativity, where there is a fixed metric and therefore a single amplitude.

7. (No-divergence condition.) Finally, as in "Evolution with respect to gravitationally interacting clocks", we assume that the wave packet of B's clock is such that no divergences occur in the denominator $1 + \lambda \hat{H}_B$. It is easy to check that the condition for no divergences implies that $\xi x_{AB} > l_P^2$, where $\xi = c\sigma$ is the characteristic with of the wave packet $\varphi_B$ in units of length, and $l_P^2 = \hbar G/c^3$ is the Planck area. The fact that the divergences occur at the Planck length is consistent with the widely-held view that, at this scale, typical quantum gravity effects become predominant.

**Knob settings and external evolution**. It is important to distinguish two different types of terms playing different roles in the constraint equations appearing in the main text. For example, consider Eq. (3), which has two different type of terms. On the one hand, Eq. (3) has the term $\hat{H}_{ext.} := \hat{H}_A + \hat{H}_B$, acting only on the clocks degrees of freedom. The clocks allow A and B to acquire information about "when" the operations take place. Roughly speaking, these degrees of freedom (together with the spatial ones[25]) correspond to the external, relative "spatiotemporal localisation of the local laboratories" in which events take place. On the other hand, the term $\hat{H}_{loc.} := \hat{f}_A(\hat{T}_A) + \hat{f}_B(\hat{T}_B)$ describes the operations on the system located at

the time defined by the clocks. Roughly speaking, they correspond to the "local operations" happening inside of the local laboratories. Motivated by the framework of process matrices[11], we make this correspondence explicit.

For the sake of generality, we consider a constraint of the form $\hat{C} = \hat{H}_{ext.} + \hat{H}_{loc.}$, where $\hat{H}_{ext.}$ is a function of the (possibly interacting) Hamiltonian of the clocks and $\hat{H}_{loc.}$ includes the interventions made on the system. All the constraints studied in this paper have this form. Dropping, for simplicity of notation, the dependence on $x, y, \ldots, z$ and $a, b, \ldots, c$, Eq. (18) reads

$$
\begin{aligned}
p &= |E|\Psi\rangle|^2 \\
&= \mathrm{Tr} E^\dagger E |\Psi\rangle \langle\Psi| \\
&= \mathrm{Tr} E^\dagger E \int d\alpha\, d\alpha'\; e^{-i\alpha(\hat{H}_{ext.} + \hat{H}_{loc.})} |\varphi\rangle \langle\varphi| e^{i\alpha'(\hat{H}_{ext.} + \hat{H}_{loc.})} \\
&= \mathrm{Tr} E^\dagger E \int d\alpha\, d\alpha'\; V(\alpha) e^{-i\alpha \hat{H}_{ext.}} |\varphi\rangle \langle\varphi| e^{i\alpha' \hat{H}_{ext.}} V^\dagger(\alpha') \\
&= \int d\alpha\, d\alpha' \mathrm{Tr} e^{-i\alpha \hat{H}_{ext.}} |\varphi\rangle \langle\varphi| e^{i\alpha' \hat{H}_{ext.}} V^\dagger(\alpha') E^\dagger E V(\alpha) \\
&= \mathrm{Tr} WM,
\end{aligned}
\tag{34}
$$

where $E$ is a measurement operator acting on the clocks and the ancillas, $W$ has matrix elements $W(\alpha, \alpha') = e^{-i\alpha \hat{H}_{ext.}} |\varphi\rangle \langle\varphi| e^{i\alpha' \hat{H}_{ext.}}$, $M$ has matrix elements $M(\alpha', \alpha) = V^\dagger(\alpha') E^\dagger E V(\alpha)$, and $V(\alpha) e^{-i\alpha \hat{H}_{ext.}} = e^{-i\alpha(\hat{H}_{ext.} + \hat{H}_{loc.})}$.

We have therefore shown that it is possible to write down the Born rule for the probability $p$, Eq. (18), in the form

$$
p = \mathrm{Tr} WM,
\tag{35}
$$

where $W$ depends only on the spatiotemporal part, corresponding to $\hat{H}_{ext.}$, and $M$ encodes the information about the operations inside the local laboratories, corresponding to $\hat{H}_{loc.}$. $M$ can depend on the spatiotemporal part as well (although this is not the case for non interacting clocks).

## Data availability
Data sharing not applicable to this article as no datasets were generated or analysed during the current study.

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

## Acknowledgements

We thank P. Allard-Guérin, L. Hardy, P. A. Hoehn, I. Kull, L. Maccone, O. Oreshkov, W. Wieland and M. P. Woods for interesting discussions. We acknowledge support from the research platform Testing Quantum and Gravity Interface with Single Photons (TURIS), the Austrian Science Fund (FWF) through the Projects No. No. I- 2906 and BeyondC (F7113-N48), and the doctoral program Complex Quantum Systems (CoQuS) under Project No. W1210-N25. We also acknowledge financial support from the EU Collaborative Project TEQ (Grant Agreement No. 766900). E.C.-R. is supported in part by the Program of Concerted Research Actions (ARC) of the Université libre de Bruxelles. A.B. is supported by H2020 through the MSCA IF pERFEcTO (Grant Agreement nr. 795782). Research at Perimeter Institute is supported in part by the Government of Canada through the Department of Innovation, Science and Economic Development Canada and by the Province of Ontario through the Ministry of Colleges and Universities. This work was funded by a grant from the Foundational Questions Institute (FQXi) Fund and the ID# 61466 grant from the John Templeton Foundation, as part of the "The Quantum Information Structure of Spacetime (QISS)" Project (qiss.fr). The opinions expressed in this publication are those of the author(s) and do not necessarily reflect the views of the John Templeton Foundation.

## Author contributions

E.C.-R., F.G., A.B. and Č.B. contributed to all aspects of the research, with the leading input of E.C.-R.

## Competing interests

The authors declare no competing interests.
