## [Peer Review File · Nature Communications]

Reviewers' Comments:

Reviewer #1:

Remarks to the Author:

In this paper, the authors investigate issue of reference frames for gravitating systems using the model of quantum clock. Starting from a timeless constrained system which represents the nature of quantum state of gravitating system, quantum clock is introduced following the idea of Page and Woiters. Then authors discussed general nature of this clock system assuming a simple structure for gravitational interaction Hamiltonian and showed how the accuracy of event depends on different two frames. In the final section, the authors consider how the causal structure of quantum mechanically suppoerposed events depends on specified frame and showed how the localization and de-localization of events appears in their model. The model and arguments adopted in this paper is very general and may be helpful if we intend to proposed laboratory experiments which reveal quantum nature of causality and quantum nature of spacetimes. I think this paper can be accepted for publication to this journal in the present form.

Reviewer #2:

Remarks to the Author:

In the present paper the authors explore how timed quantum measurements/operations involving different actors/laboratories can be described in an extension of the standard formulation of quantum theory when the laboratories (in terms of quantum clocks they contain) are in gravitational interaction with each other (Section IV) or with an external source (Section V). Crucially, the gravitational interaction is allowed to depend on the (quantum) states of the laboratories (more specifically the clocks) and/or that of the external source. In particular, effectively superpositions of different spacetime metrics are involved. To this end the authors develop a notion of "time reference frames" (Section III) arising from a fusion of what they call "the timeless approach" to quantum theory (Section II), mostly ref. [15], and the notion of "quantum reference frame". The authors are able to model an (extremely simplified) quantum gravitational interaction between the (clocks in the) laboratories as well as the influence of an external gravitational mass in superposition of different spatial positions. While the system as a whole is described in terms of a "history state", the notion of "time reference frame" refers to a reduction of the history state with respect to the clock in a chosen laboratory. The authors show that while in each of these "time reference frames" the time of an operation effected on a joint target system by the associated laboratory is well defined, the time other laboratories apply operations need not be well defined. Still, evolution within the associated laboratory appears unitary. Moreover, the authors show how transformations between different "time reference frames" can be performed.

There are a number of shortcomings of the paper. Firstly and most fundamentally the "timeless" approach to quantum theory (briefly introduced in Section II) is still far from being a well-developed or coherent framework to describe quantum theory in a timeless context. Moreover, the references [14,15,17,18] given cannot be considered to describe exactly the same approach. These approaches are still very much based on a non-relativistic view of spacetime, although incorporating a certain relaxation of an absolute notion of time, but still keeping time completely separate from space. To bring these approaches closer to their stated goal a first step (much before moving to a general relativistic context) would be to try to make them compatible with special relativity, i.e., with relativistic quantum field theory.

Related to this point, the modeling of the spatiotemporal properties of the different component systems considered in the paper is extremely simplified and almost completely non-relativistic. Improving this substantially would likely require a much more powerful underlying framework for quantum theory for a context without fixed metric. While parts of this exist in the literature, making this work would certainly be a substantial task clearly beyond the scope of the present

paper.

Another weakness of the paper is a serious lack of mathematical rigor. Indeed, I think it is fair to say that most mathematical expressions in the paper are not well defined. With some effort I think the authors could have done better. For example, replacing continuous time by a discretization (as an approximation) could have helped quite a bit. However, as the interested reader can mostly imagine how statements could be made rigorous I will refrain from asking the authors for changes in this direction. However, I would like to urge the authors to pay more attention to mathematical well definedness in future works.

In summary, given that bringing together quantum theory and gravity is such a difficult endeavor and in view of the interesting and compelling (if tentative) results presented by the authors in this direction I think the merits of the paper clearly outweigh the criticism. I am also convinced the paper will find broad interest in the community. Therefore I recommend publication in Nature Communications.

There are just two points that I would like the authors to address before publication:

- It is somewhat disingenuous to talk about "formulations of quantum mechanics such as..." when you need to refer to a specific one as a reference for the presentation in Section II. The four different references given [14,15,17,18] are partially contradictory and do not all describe the same framework. However, It seems that [15] is closest to what is actually used. Thus a reference to [15] as definitive for the present article should be made. The others could be mentioned as being predecessors [14] or somewhat similar [17,18].
- The expression "indefinite metric" is often used and clearly meant in analogy to "indefinite causal structure". I.e. the intended meaning is "absence of a well defined metric". However, "indefinite metric" has also a (completely different) meaning as a mathematical term. Perhaps a footnote early on in the paper to avoid confusion would be in order.

Reviewer #3:

Remarks to the Author:

The authors construct a framework for temporal reference frames which associates a quantum clock to each time reference frame. They argue that the impossibility to find a reference frame in which all events are localised is a signature of an indefinite metric, which might yield an indefinite causal order of events.

These results are of interest and correct as far as I can tell.

However, their statements on the signification of the present manuscript is more or less bombastic. We know that a broadly accepted theory of quantum gravity is lacking because it is not easy to understand the consequences of replacing gravitating classical matter by gravitating quantum systems in general relativity. In the third paragraph of the introduction they authors listed three well known difficulties in how to make quantum mechanical predictions when quantum systems act as gravitational. However, I do not see any actual progress on the mentioned three difficulties in the performed framework. Therefore, the mere study on the temporal localisation of events with respect to different time reference frames does not break new ground to the extent that should be expected in Nature Communications. I also do not see significant applications or important physical implications of their result. A specialized journal, perhaps npj Quantum Information, would be a more appropriate venue for the paper.

In addition, my main concern for the framework of the "time reference frames" is on the description of quantum measurements performed at multiple times in the "timeless" framework. We know that the standard formulation of quantum measure theory relies on a fixed space-time

metric. That is to say, a physical realisation of any measurement performed on a quantum system relies on a definite space-time metric, which are crucial for defining the basis of and outcome of the measurements. If the space-time metric is indefinite, the processing of quantum measurement is also indefinite. However, multiple measurements is important for the definition of the "timeless" approach. The authors should clarify this point in the manuscript.

We thank Reviewer #2 and Reviewer #3 for their comments. In the following we give a point-by-point reply to their remarks and mention the changes made to the manuscript in the light of their comments.

Reviewer #2 (Remarks to the Author):

In the present paper the authors explore how timed quantum measurements/operations involving different actors/laboratories can be described in an extension of the standard formulation of quantum theory when the laboratories (in terms of quantum clocks they contain) are in gravitational interaction with each other (Section IV) or with an external source (Section V). Crucially, the gravitational interaction is allowed to depend on the (quantum) states of the laboratories (more specifically the clocks) and/or that of the external source. In particular, effectively superpositions of different spacetime metrics are involved. To this end the authors develop a notion of "time reference frames" (Section III) arising from a fusion of what they call "the timeless approach" to quantum theory (Section II), mostly ref. [15], and the notion of "quantum reference frame". The authors are able to model an (extremely simplified) quantum gravitational interaction between the (clocks in the) laboratories as well as the influence of an external gravitational mass in superposition of different spatial positions. While the system as a whole is described in terms of a "history state", the notion of "time reference frame" refers to a reduction of the history state with respect to the clock in a chosen laboratory. The authors show that while in each of these "time reference frames" the time of an operation effected on a joint target system by the associated laboratory is well defined, the time other laboratories apply operations need not be well defined. Still, evolution within the associated laboratory appears unitary. Moreover, the authors show how transformations between different "time reference frames" can be performed.

There are a number of shortcomings of the paper. Firstly and most fundamentally the "timeless" approach to quantum theory (briefly introduced in Section II) is still far from being a well-developed or coherent framework to describe quantum theory in a timeless context. Moreover, the references [14,15,17,18] given cannot be considered to describe exactly the same approach. These approaches are still very much based on a non-relativistic view of spacetime, although incorporating a certain relaxation of an absolute notion of time, but still keeping time completely separate from space. To bring these approaches closer to their stated goal a first step (much before moving to a general relativistic context) would be to try to make them compatible with special relativity, i.e., with relativistic quantum field theory.

We agree with Reviewer #2 that constructing a special relativistic extension of the timeless framework in which time and space appear on equal footing is an important endeavour. In fact, the idea of a "quantum reference frame" can be a valuable ingredient towards such a goal. For this reason, at the end of the third paragraph in page 3 of the new version, we mention Ref. 25 as a method for treating the case of spatial (non-relativistic) quantum reference frames, which complements the treatment of temporal reference frames presented in this work. We believe that putting together relativistic extensions of these two frameworks would provide significant progress towards a fully relativistic timeless framework. In fact, the recent publication "Relativistic quantum reference frames: the operational meaning of spin, Phys. Rev. Lett. (2019)" already contains a special relativistic treatment of spatial reference frames. In the light of this comment by Reviewer #2, in the Discussion section we note the significance of building a special relativistic timeless approach, and cite the mentioned reference as a contribution towards this goal.

On the other hand, we believe that the significance of our paper lies in the construction of a framework for studying situations with an indefinite space-time metric rather than studying situations with relativistic symmetry on a fixed metric. As Reviewer #2 correctly points out, the construction of a framework for relativistic quantum fields on an indefinite spacetime metric is outside the scope of our work.

Related to this point, the modeling of the spatiotemporal properties of the different component systems considered in the paper is extremely simplified and almost completely non-relativistic. Improving this substantially would likely require a much more powerful underlying framework for quantum theory for a context without fixed metric. While parts of this exist in the literature, making this work would certainly be a substantial task clearly beyond the scope of the present paper.

We thank Reviewer #2 for this comment. We would like to point out that an important feature of the physical situations analysed in our manuscript is relativistic time dilation due to the presence of gravitating objects. Therefore, in this sense, our work contains truly relativistic elements. On the other hand, we agree that constructing a framework in which special relativistic symmetry is fully manifest is a significant task.

Related to this point, in the last paragraph of Supplementary Note 2, we sketch an idea on how to construct a more complete (and relativistic) treatment of the gravitational quantum switch of Ref. 12. We propose to construct K operators (the ones that define an event via the coupling of a system with an ancilla) that commute outside the light cone and do not commute otherwise. While we believe this idea is implementable in principle, its development is outside the scope of the present work.

Another weakness of the paper is a serious lack of mathematical rigor. Indeed, I think it is fair to say that most mathematical expressions in the paper are not well defined. With some effort I think the authors could have done better. For example, replacing continuous time by a discretization (as an approximation) could have helped quite a bit. However, as the interested reader can mostly imagine how statements could be made rigorous I will refrain from asking the authors for changes in this direction. However, I would like to urge the authors to pay more attention to mathematical well definedness in future works.

We thank Reviewer #2 for this comment, and coincide with their opinion that replacing continuous time by a discretisation is a good idea in order to define the mathematical objects of this paper in a fully rigorous way.

On the other hand, we would like to point out that the mathematics used in our manuscript are sufficient for establishing the physical consequences reported in our work.

In summary, given that bringing together quantum theory and gravity is such a difficult endeavor and in view of the interesting and compelling (if tentative) results presented by

the authors in this direction I think the merits of the paper clearly outweigh the criticism. I am also convinced the paper will find broad interest in the community. Therefore I recommend publication in Nature Communications.

We thank Reviewer #2 for their positive opinion of our results and for their recommendation to publish our paper.

There are just two points that I would like the authors to address before publication:

- It is somewhat disingenuous to talk about "formulations of quantum mechanics such as..." when you need to refer to a specific one as a reference for the presentation in Section II. The four different references given [14,15,17,18] are partially contradictory and do not all describe the same framework. However, It seems that [15] is closest to what is actually used. Thus a reference to [15] as definitive for the present article should be made. The others could be mentioned as being predecessors [14] or somewhat similar [17,18].

In view of this comment by Reviewer #2, we have:

1) Modified the Results section and included, in subsection "Reference frames for events and time evolution," the following sentence: "To formulate this idea, we consider as a starting point the timeless approach to quantum mechanics [20, 21, 28], in particular the version of Ref. [21]." We believe this modification addresses Reviewer #2's concern and acknowledges Ref. 21 (Ref. 15 of the previous version) as the closest to what we use in our work.

2) Modified the Methods section and included, in subsection "Review of the timeless approach to quantum mechanics" the following sentence: "Let us briefly review the "timeless" approach to quantum mechanics. This approach was proposed by Page and Wootters [20] and further developed by Giovanetti, Lloyd and Maccone [21]. Our review follows closely this latter version, which is particularly relevant to our work. A similar approach is developed by covariant quantum mechanics [23, 24]." (Note the following change in the numbering of the references with respect to the previous version: Ref. 14 → Ref. 20; Ref. 17 → Ref. 23; Ref. 18 → Ref. 24.)

- The expression "indefinite metric" is often used and clearly meant in analogy to "indefinite causal structure". I.e. the intended meaning is "absence of a well defined metric". However, "indefinite metric" has also a (completely different) meaning as a mathematical term. Perhaps a footnote early on in the paper to avoid confusion would be in order.

We thank reviewer #2 for pointing this out. In order to avoid confusion, in the Introduction of the new version we have added the following clarification: "... indefinite spacetime metric, that is, a metric whose values are not given a priori, independently of any observations carried out on quantum systems." We believe this sentence explains unambiguously what we mean by an "indefinite metric."

Reviewer #3 (Remarks to the Author):

The authors construct a framework for temporal reference frames which associates a quantum clock to each time reference frame. They argue that the impossibility to find a reference frame in which all events are localised is a signature of an indefinite metric, which might yield an indefinite causal order of events.

These results are of interest and correct as far as I can tell.

We thank Reviewer #3 for finding our results interesting.

However, their statements on the signification of the present manuscript is more or less bombastic. We know that a broadly accepted theory of quantum gravity is lacking because it is not easy to understand the consequences of replacing gravitating classical matter by gravitating quantum systems in general relativity. In the third paragraph of the introduction they authors listed three well known difficulties in how to make quantum mechanical predictions when quantum systems act as gravitational. However, I do not see any actual progress on the mentioned three difficulties in the performed framework. Therefore, the mere study on the temporal localisation of events with respect to different time reference frames does not break new ground to the extent that should be expected in Nature Communications. I also do not see significant applications or important physical implications of their result. A specialized journal, perhaps npj Quantum Information, would be a more appropriate venue for the paper.

We thank Reviewer #3 for this comment. However, we kindly disagree with their assessment, and we do believe our work breaks new grounds towards understanding the physics of gravitating quantum systems. The main reasons for this belief are summarised in the following points:

- 1) We establish a mathematical framework suitable for the study of physical events and dynamical laws from the point of view of multiple quantum temporal reference frames. We believe this is a significant achievement, potentially valuable to our understanding of gravitating quantum matter. Indeed, understanding how physical quantities and entities transform under changes of reference frames has been crucial for the development of modern theoretical physics. Moreover, exploring the quantum features of such reference frames is essential for comprehending what happens when quantum matter gravitates. Therefore, the establishment of such mathematical framework is, in itself, an important contribution.
- 2) After establishing our mathematical framework, we use it to study the fundamental concept of “event.” We do so in cases where both the quantum and the gravitational properties of physical systems are relevant, leading to an indefinite space-time metric. We believe shedding light on the concept of event under these circumstances is a very important task. In this respect, we find that the temporal localisability of an event becomes a relative notion, depending on the temporal reference frame. This is contrary to the case where the metric is definite, and points to a new way of understanding the concept of “event”. In view of the fundamental status of this concept, we believe our work might be important to an operational understanding of space-time in the presence of gravitating quantum matter. As a matter of fact, we have little clue as to how our most basic notions of physics will be modified, if at all, in a quantum theory of gravity. Our work breaks new ground in the sense that it proposes one such modification –regarding the concept of “event”. This modification is far from a

“physical curiosity”. Rather, it provides an important conceptual insight into the physics of gravitating quantum systems, which could contribute to new ideas regarding the problem of quantum gravity.

- 3) We demonstrate that, under our definition of event (as a quantum operation triggered by a quantum clock), there exists a reference frame in which that event is localised in time, as defined by the clock corresponding to that frame. Moreover, we show that the mathematical description of this event (quantum operation) acquires its usual, quantum mechanical form when “jumping” into that frame. More precisely, it acquires the form of a dilation of a quantum operation. This means that, in that frame and for that event, we can “ignore” the fact that there is no definite space-time metric, because the measurement is described by ordinary quantum mechanics. In our view, this is an important achievement, because it manages to reduce a situation which we, in principle, would not know how to describe formally (a quantum operation in presence of gravitating quantum systems), into a situation which we know how to handle (an ordinary dilation of a quantum operation controlled by a quantum clock).

In addition, my main concern for the framework of the “time reference frames” is on the description of quantum measurements performed at multiple times in the “timeless” framework. We know that the standard formulation of quantum measure theory relies on a fixed space-time metric. That is to say, a physical realisation of any measurement performed on a quantum system relies on a definite space-time metric, which are crucial for defining the basis of and outcome of the measurements. If the space-time metric is indefinite, the processing of quantum measurement is also indefinite. However, multiple measurements is important for the definition of the “timeless” approach. The authors should clarify this point in the manuscript.

We thank Reviewer #3 for raising this point. We believe our answer 3) above addresses this point to a significant extent. The idea is the following: Even in an indefinite metric we can find a reference frame where measurements are described by ordinary dilations of quantum operations (controlled by a quantum clock). Therefore, in this frame we can reduce the problem to a problem which we already know how to solve. Note that, in our framework, nothing prevents us from defining multiple measurements in the same way, so we believe that multiple measurements are not problematic in this respect.

Admittedly, as we point out in the Discussion section, there is still important work to be done in order to fully describe quantum measurements on indefinite causal structures (indefinite metrics). For example, accounting fully for special relativistic effects, or building a framework which is amenable to a field theoretic description are important tasks. However, we believe the results of this manuscript make an important step and break new ground in this direction.